# RETHINKING RESIDUAL ERRORS IN COMPENSATION-BASED LLM QUANTIZATION

Shuaiting Li[1,2][*], Juncan Deng[1,2][*], Kedong Xu[2], Rongtao Deng[2], Hong Gu[2], Minghan Jiang[1], Haibin Shen[1][†], Kejie Huang[1][†]

[1]Zhejiang University      [2]vivo Mobile Communication Co., Ltd

{list,dengjuncan,jiangmh,shen_hb,huangkejie}@zju.edu.cn

{xukedong, dengrongtao, guhong}@vivo.com

## ABSTRACT

Methods based on weight compensation, which iteratively apply quantization and weight compensation to minimize the output error, have recently demonstrated remarkable success in quantizing Large Language Models (LLMs). The representative work, GPTQ, introduces several key techniques that make such iterative methods practical for LLMs with billions of parameters. GPTAQ extends this approach by introducing an asymmetric calibration process that aligns the output of each quantized layer with its full-precision counterpart, incorporating a residual error into the weight compensation framework. In this work, we revisit the formulation of the residual error. We identify a sub-optimal calibration objective in existing methods: during the intra-layer calibration process, they align the quantized output with the output from compensated weights, rather than the true output from the original full-precision model. Therefore, we redefine the objective to precisely align the quantized model's output with the original output of the full-precision model at each step. We then reveal that the residual error originates not only from the output difference of the preceding layer but also from the discrepancy between the compensated and original weights within each layer, which we name the 'compensation-aware error'. By inheriting the neuron decomposition technique from GPTAQ, we can efficiently incorporate this compensation-aware error into the weight update process. Extensive experiments on various LLMs and quantization settings demonstrate that our proposed enhancements integrate seamlessly with both GPTQ and GPTAQ, significantly improving their quantization performance. Our code is publicly available at https://github.com/list0830/ResComp.

## 1 INTRODUCTION

Since the advent of the Transformer architecture (Vaswani, 2017), it has become the cornerstone of Large Language Models (LLMs) (Touvron et al., 2023; DeepSeek-AI, 2024; Team, 2023; OpenAI, 2022; xAI, 2024; et al., 2023), driving unprecedented growth in their scale (Brown et al., 2020). This scaling is primarily driven by two of its core design principles: first, the self-attention mechanism, whose computational complexity grows quadratically with the input sequence length; and second, the massive feed-forward networks (FFNs) within each Transformer block, which constitute the vast majority of the model's parameters. While these designs have led to remarkable performance improvements, they have also resulted in enormous parameter counts and computational requirements. For instance, the Llama 3 70B model (Meta, 2024) comprises 70 billion parameters and requires 140GB of GPU memory for inference at its native 16-bit precision. Such immense resource demands severely restrict the deployment of these state-of-the-art models in resource-constrained environments, thereby underscoring the critical importance of model compression techniques such as quantization.

---

[*]Equal contribution, Work done during an internship at vivo Mobile Communication.
[†]Corresponding Authors

Quantization is a classic and effective model compression technique (Gholami et al., 2022) that significantly reduces memory footprint and accelerates computation by mapping high-precision floating-point weights and activations to low-precision integer formats, all without altering the model architecture. Quantization methods are broadly categorized into two families: Quantization-Aware Training (QAT) (Jacob et al., 2018; Shao et al., 2023; Liu et al., 2024) and Post-Training Quantization (PTQ) (Lin et al., 2023; Dettmers et al., 2022; Frantar & Alistarh, 2022). QAT typically achieves higher accuracy by updating quantization parameters through gradient-based optimization during a fine-tuning phase. However, its substantial fine-tuning cost renders it impractical for today's ultra-large models. In contrast, PTQ requires no fine-tuning, enabling model quantization at a remarkably low cost. This efficiency establishes PTQ as the dominant paradigm for LLM compression (Nagel et al., 2021). GPTQ (Frantar et al., 2022) is a representative PTQ approach that performs layer-wise calibration and leverages Hessian information to compensate for quantization errors. Building upon this, GPTAQ (Li et al., 2025) introduces an asymmetric calibration process that effectively mitigates the layer-by-layer accumulation of errors. The core idea of this process is to align the output of each quantized layer with its full-precision counterpart by propagating the output error from the preceding layer into the current layer's calibration as a residual.

In this work, we focus on the output alignment problem within the intra-layer iterative process of the GPTQ and GPTAQ methods. We find that previous works compute the calibration target at each step using the compensated weights, thereby neglecting alignment with the original output of the full-precision layer, which we argue is the more precise calibration objective. We reformulate this new objective and reveal that a more precise residual error should not only include the output error propagated from the preceding layer but also an intrinsic error introduced by weight compensation within the current layer, which we term the compensation-aware error. We leverage the neuron decomposition technique from GPTAQ to efficiently incorporate this compensation-aware error into the weight update process. Furthermore, our proposed improvements readily integrate with both the GPTQ and GPTAQ frameworks. The main contributions of this paper are summarized as follows:

- We provide an in-depth analysis of the output alignment at each step and propose a more precise residual error formulation that incorporates both the inter-layer propagated output error and the intra-layer error introduced by weight compensation.

- We develop an efficient method to compute the proposed compensation-aware error and incorporate it into the weight update process, building upon the neuron decomposition.

- Our method requires only minimal modifications to the GPTQ and GPTAQ frameworks and achieves significant performance improvements across a wide range of large language models and quantization settings.

## 2 RELATED WORK

**Post-Training Quantization for LLMs.** PTQ is a vital technique for compressing large models without retraining (Gholami et al., 2022), typically falling into two categories. The first category mitigates quantization difficulty by redistributing weights or activations. To handle outliers, methods like LLM.int8() (Dettmers et al., 2022) use mixed-precision, while others apply activation smoothing (Xiao et al., 2023), channel reordering (Yuan et al., 2023), or global orthogonal transforms (e.g., Hadamard matrices) to unify distributions (Tseng et al., 2024; Ashkboos et al., 2024; Liu et al., 2024). The second category directly compensates for quantization error. For instance, GPTQ (Frantar et al., 2022) minimizes layer-wise output error by using second-order information to iteratively update remaining full-precision weights.

**Compensation-based Quantization.** The origins of compensation-based quantization can be traced back to early pioneering work on network pruning (LeCun et al., 1989; Hassibi et al., 1993). The Optimal Brain Damage (OBD) method (LeCun et al., 1989) introduces a framework for pruning by leveraging second-order information under the assumption that the Hessian matrix is diagonal. Subsequently, the Optimal Brain Surgeon (OBS) framework (Hassibi et al., 1993) improves upon this by relaxing the diagonal Hessian assumption, which allows for more accurate weight updates. Optimal Brain Quantization (OBQ) (Frantar & Alistarh, 2022) generalizes this second-order pruning framework to the task of quantization. OBQ processes weights sequentially based on the magnitude of the quantization error, continuously adjusting the remaining full-precision weights to compensate. However, its direct application to LLMs remains computationally challenging. To address the

computational challenges of OBQ, GPTQ (Frantar et al., 2022) achieves significant efficiency gains through techniques like lazy batch-updates and a Cholesky reformulation, enabling practical calibration on large-scale models. More recently, GPTAQ (Li et al., 2025) extends the GPTQ framework by introducing an asymmetric calibration mechanism to address error accumulation. This asymmetric calibration ensures that the output of each quantized layer aligns with that of its full-precision counterpart. Our work builds upon the GPTQ and GPTAQ framework, offering an in-depth analysis and further enhancing its error compensation mechanism.

## 3 BACKGROUND

Standard compensation-based quantization methods typically formulate the problem as an iterative optimization process. As shown in the upper part of Fig. 1, the procedure sequentially quantizes weight columns and compensates for the quantization error by adjusting the remaining unquantized weights. We briefly review the formulations of GPTQ and GPTAQ below to establish the baseline for our method.

### 3.1 NOTATIONS

We adopt the problem formulation and notation from GPTAQ Li et al. (2025) solely to facilitate direct comparison. Throughout this paper, we use lowercase boldface (e.g., $\mathbf{w}$) for vectors and uppercase boldface (e.g., $\mathbf{W}$) for matrices. We focus on the standard linear layer computation $\mathbf{y} = \mathbf{w}\mathbf{X}$, where the weight row-vector $\mathbf{w} \in \mathbb{R}^{1 \times n}$ multiplies the input activation matrix $\mathbf{X} \in \mathbb{R}^{n \times k}$ to produce the output $\mathbf{y} \in \mathbb{R}^{1 \times k}$. We denote the quantized version of weights as $\hat{\mathbf{w}}$. Furthermore, we use the subscript notation $\mathbf{H}_{-q}$ to represent the matrix $\mathbf{H}$ with its $q$-th row excluded.

Assuming quantization is performed in column order, We define $\mathbf{w}^{(q)} \in \mathbb{R}^{1 \times n}$ as the weight state after $q$ steps of quantization and compensation, with its first $q$ columns (from 0 to $q - 1$) already quantized. $\mathbf{w}^{(0)}$ represents the original floating-point weight. Two computation flows are defined to differentiate input sources: As shown in Eqn. 1, When calculating the input for the $l$-th layer, the *Quant-flow* $\mathbf{X}$ aims to simulate the quantized inference process by using the previously quantized layers for computation. Conversely, the *FP-flow* $\widetilde{\mathbf{X}}$ consistently uses the original floating-point layers for computation.

$$\widetilde{\mathbf{X}}^l = F(\mathbf{w}^{l-1}, \widetilde{\mathbf{X}}^{l-1}), \quad \mathbf{X}^l = F(\widehat{\mathbf{w}}^{l-1}, \mathbf{X}^{l-1}) \tag{1}$$

### 3.2 OBQ & GPTQ

When quantizing the $l$-th layer, OBQ and GPTQ employs the *Quant-flow* throughout the entire process. The objective of both OBQ Frantar & Alistarh (2022) and GPTQ Frantar et al. (2022) is to minimize the reconstruction error between the floating-point weights and the quantized weights under the same input. Therefore, the **high-level (layer-level)** optimization objective is defined as:

$$\min_{\widehat{\mathbf{w}}} ||\widehat{\mathbf{w}}\mathbf{X} - \mathbf{w}\mathbf{X}||_F^2, \tag{2}$$

When quantizing the $q$-th column, to derive the optimal update $\Delta\mathbf{w}$ for remaining weights, the **low-level (column-level)** objective is formulated as:

$$\min_{\Delta\mathbf{w}} ||(\mathbf{w}_{q:}^{(q)} + \Delta\mathbf{w})\mathbf{X}_{q:,:} - \mathbf{w}_{q:}^{(q)}\mathbf{X}_{q:,:}||_F^2, \text{s.t. } \Delta\mathbf{w}_q = \hat{\mathbf{w}}_q^{(q)} - \mathbf{w}_q^{(q)} \tag{3}$$

The above equation can be solved for the corresponding $\Delta\mathbf{w}$ using the Lagrange multiplier method.

$$q = \arg\min_q \frac{(\hat{\mathbf{w}}_q - \mathbf{w}_q)^2}{\mathbf{H}_{qq}^{-1}}, \quad \Delta\mathbf{w} = \frac{(\hat{\mathbf{w}}_q - \mathbf{w}_q)}{\mathbf{H}_{qq}^{-1}} \cdot (\mathbf{H}_{q,:}^{-1}), \tag{4}$$

where $\mathbf{H}^{-1} = (\mathbf{X}\mathbf{X}^\top)^{-1}$ represents the inverse Hessian matrix. Since the quantized weights will no longer be updated, the inverse Hessian for the remaining weights should be updated via gaussian elimination. However, applying OBQ to large models encounters severe efficiency issues. Firstly, it requires storing a different inverse Hessian for every weight row, and secondly, it necessitates

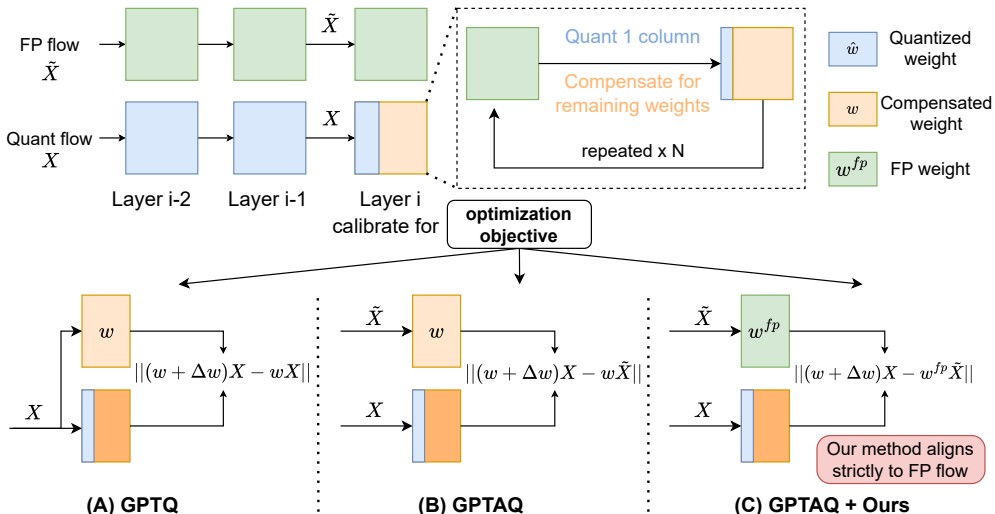

Figure 1: Overview of compensation-based LLM quantization methods. (A) **GPTQ**: Minimizes the layer-wise reconstruction error based on the current quantized input flow, neglecting error accumulation from previous layers. (B) **GPTAQ**: Introduces asymmetric calibration to address inter-layer error accumulation, but uses the compensated weights as the calibration target during the update process. (C) **GPTAQ+Ours**: Strictly aligns with the original full-precision output by incorporating a novel "Compensation-Aware Error," which corrects the discrepancy between compensated and original weights.

recalculating the inverse Hessian at every iteration. To address these problems, GPTQ first proposed using the same quantization order for all rows to share a single Hessian matrix. It then introduced the Cholesky reformulation $\mathbf{H}^{-1} = \mathbf{L}\mathbf{L}^T$ to pre-calculate the required inverse Hessian information for each step, thus avoiding repeated computations. This breakthrough made efficient compensation-based quantization possible on large-scale models.

### 3.3 GPTAQ

GPTAQ Li et al. (2025) identifies a key drawback in the former method: the exclusive use of the *Quant-flow* for calibration. Due to the accumulation of errors across successive layers, the calibration baseline $\mathbf{w}\mathbf{X}$ itself becomes biased, whereas the output of each layer in the *FP-flow*, $\mathbf{w}\widetilde{\mathbf{X}}$, serves as the accurate reference. Therefore, GPTAQ simultaneously considers both $\mathbf{X}$ and $\widetilde{\mathbf{X}}$, attempting to align the *Quant-flow* and *FP-flow* at every layer. Therefore, the **high-level (layer-level)** optimization objective is defined as:

$$\min_{\widehat{\mathbf{w}}} ||\widehat{\mathbf{w}}\mathbf{X} - \mathbf{w}\widetilde{\mathbf{X}}||_F^2, \tag{5}$$

When quantizing the $q$-th column, The **low-level (column-level)** objective in GPTAQ is then formulated as:

$$\min_{\Delta\mathbf{w}} ||(\mathbf{w}_{q:}^{(q)} + \Delta\mathbf{w})\mathbf{X}_{q:,:} - \mathbf{w}_{q:}^{(q)}\widetilde{\mathbf{X}}_{q:,:}||_F^2, \text{s.t. } \Delta\mathbf{w}_q = \hat{\mathbf{w}}_q^{(q)} - \mathbf{w}_q^{(q)} \tag{6}$$

To maintain consistency with the solving procedure in GPTQ, GPTAQ introduces the concept of **residual error**, defined as $\mathbf{r} = \mathbf{w}_{q:}^{(q)}\widetilde{\mathbf{X}}_{q:,:} - \mathbf{w}_{q:}^{(q)}\mathbf{X}_{q:,:}$. The **column-level** objective becomes:

$$\min_{\Delta\mathbf{w}} ||\Delta\mathbf{w}\mathbf{X}_{q:,:} - \mathbf{r}_q||_F^2, \text{s.t. } \Delta\mathbf{w}_q = \hat{\mathbf{w}}_q^{(q)} - \mathbf{w}_q^{(q)} \tag{7}$$

The solution to this Lagrangian introduces an additional correction term to the standard update:

$$\Delta\mathbf{w} = \underbrace{\frac{(\hat{\mathbf{w}}_q - \mathbf{w}_q)}{\mathbf{H}_{qq}^{-1}} \cdot (\mathbf{H}_{q,:}^{-1})}_{\text{Standard Update}} + \underbrace{\mathbf{r}\mathbf{X}^\top \mathbf{H}_{-q}^{-1}}_{\text{Error Correction}} . \tag{8}$$

## 4 METHODS

### 4.1 RETHINKING RESIDUAL ERRORS

The high-level (layer-level) optimization objective of GPTAQ is exactly correct, as it consistently uses the output of the *FP-flow*, $\mathbf{w}\widetilde{\mathbf{X}}$, as the calibration target (or reference). However, during the iterative quantization process, we found that its column-level optimization objective deviates. Let us re-examine the objective of GPTAQ when quantizing the $q$-th column:

$$\min_{\Delta \mathbf{w}} ||(\mathbf{w}_{q:}^{(q)} + \Delta \mathbf{w})\mathbf{X}_{q:,:} - \mathbf{w}_{q:}^{(q)}\widetilde{\mathbf{X}}_{q:,:}||_F^2, \text{s.t. } \Delta \mathbf{w}_q = \hat{\mathbf{w}}_q^{(q)} - \mathbf{w}_q^{(q)} \tag{9}$$

While this formulation holds for the initial step, as $\mathbf{w}^{(0)}\widetilde{\mathbf{X}}$ is precisely the output from the floating-point layer. However, this becomes inaccurate in subsequent iterations because for $q \geq 1$, $\mathbf{w}^{(q)}\widetilde{\mathbf{X}}$ no longer represents the accurate output. To strictly maintain output alignment with $\mathbf{w}^{(0)}\widetilde{\mathbf{X}}$ at every step, we can formulate the objective function as:

$$\min_{\Delta \mathbf{w}} ||(\mathbf{w}^{(q)} + \Delta \mathbf{w})\mathbf{X} - \mathbf{w}^{(0)}\widetilde{\mathbf{X}}||_F^2$$
$$\text{s.t. } \Delta \mathbf{w}_{:q} = 0, \Delta \mathbf{w}_q = \hat{\mathbf{w}}_q^{(q)} - \mathbf{w}_q^{(q)} \tag{10}$$

where the left and right sides of the equation represent the complete outputs of the quantized layer and the full-precision layer, respectively. The update term $\Delta w$ is still only applied to non-quantized columns. Eqn. 10 is equal to:

$$\min ||\mathbf{w}^{(q)}\mathbf{X} + \Delta \mathbf{w}_q:\mathbf{X}_{q:,:} - \mathbf{w}^{(0)}\widetilde{\mathbf{X}}||_F^2, \text{s.t. } \Delta \mathbf{w}_q = \hat{\mathbf{w}}_q^{(q)} - \mathbf{w}_q^{(q)} \tag{11}$$

To align with the notation used in GPTAQ, we can further express the preceding equation as:

$$\min ||\Delta \mathbf{w}_q:\mathbf{X}_{q:,:} - (\mathbf{w}^{(0)}\widetilde{\mathbf{X}} - \mathbf{w}^{(q)}\mathbf{X})||_F^2, \text{s.t. } \Delta \mathbf{w}_q = \hat{\mathbf{w}}_q^{(q)} - \mathbf{w}_q^{(q)} \tag{12}$$

Therefore, $\mathbf{r}' = (\mathbf{w}^{(0)}\widetilde{\mathbf{X}} - \mathbf{w}^{(q)}\mathbf{X})$ is the new residual error, and can be written as:

$$\mathbf{r}' = \underbrace{\mathbf{w}^{(q)}(\widetilde{\mathbf{X}} - \mathbf{X})}_{r1} + \underbrace{(\mathbf{w}^{(0)} - \mathbf{w}^{(q)})\widetilde{\mathbf{X}}}_{r2} \tag{13}$$

The first term, $\mathbf{r}1$, represents the residual error arising from the input error, which is adopted in GPTAQ. The new second term, $\mathbf{r}2$, represents the error introduced by the intra-layer weight compensation, which we name the **'Compensation-aware Error'**. Therefore, when quantizing the q-th column, by replacing $\mathbf{r}$ with $\mathbf{r}'$ in Eqn. 8, we can derive the new $\Delta \mathbf{w}$ as:

$$\Delta \mathbf{w} = \frac{(\hat{\mathbf{w}}_q^{(q)} - \mathbf{w}_q^{(q)})}{\mathbf{H}_{qq}^{-1}} \cdot (\mathbf{H}_{q,:}^{-1}) + (\mathbf{r}1 + \mathbf{r}2)\mathbf{X}^\top \mathbf{H}_{-q}^{-1}, \tag{14}$$

### 4.2 EFFICIENT SOLUTION

Following GPTQ, we process weights in an arbitrary order, which is sequentially from the first to the last, to enable parallel processing of all rows. For each column $q = 0, 1, ...n - 1$, we compute the weight update $\Delta \mathbf{W}$ across all rows with:

$$\Delta \mathbf{W} = \frac{(\widehat{\mathbf{W}}_{:,q}^{(q)} - \mathbf{W}_{:,q}^{(q)})}{\mathbf{H}_{qq}^{-1}} \cdot (\mathbf{H}_{q,:}^{-1}) + (\mathbf{R}1 + \mathbf{R}2)\mathbf{X}^\top \mathbf{H}_{-q}^{-1} \tag{15}$$

where $\mathbf{R}1 = \mathbf{W}^{(q)}\widetilde{\mathbf{X}} - \mathbf{W}^{(q)}\mathbf{X}$, $\mathbf{R}2 = (\mathbf{W}^{(0)} - \mathbf{W}^{(q)})\widetilde{\mathbf{X}}$. However, re-computation of $\mathbf{R}1 \& \mathbf{R}2$ is quite heavy for large foundation models, as pointed out by GPTAQ. alternatively, GPTAQ proposes an efficient neuron decomposition technique to $\mathbf{R}1$. Denote $\Delta \mathbf{X} = \widetilde{\mathbf{X}} - \mathbf{X}$. Similarly, we apply neuron decomposition to $\mathbf{R}2$.

$$\mathbf{R}1 = \mathbf{W}^{(q)}\Delta \mathbf{X} = \sum_{q=0}^{n-1} \mathbf{W}_{:,q}^{(q)}\Delta \mathbf{X}_{q,:} \tag{16}$$

$$\mathbf{R}2 = (\mathbf{W}^{(0)} - \mathbf{W}^{(q)})\widetilde{\mathbf{X}} = \sum_{q=0}^{n-1} (\mathbf{W}_{:,q}^{(0)} - \mathbf{W}_{:,q}^{(q)})\widetilde{\mathbf{X}}_{q,:} \tag{17}$$

---

**Algorithm 1** GPTAQ quantization with Compensation-aware Error (CAE)

---

1: **Input:** FP weight $\mathbf{W}$, calibration input $\mathbf{X}$, FP input $\tilde{\mathbf{X}}$, and Block size $B$
2: $W^{(0)} \leftarrow W, \mathbf{H} \leftarrow \mathbf{X}\mathbf{X}^\top, \Delta\mathbf{X}\mathbf{X}^\top \leftarrow (\tilde{\mathbf{X}} - \mathbf{X})\mathbf{X}^\top, \mathbf{L} = Inverse\_Cholesky(\mathbf{H} + \lambda_1\mathbf{I})$
3: $\mathbf{P}1 \leftarrow \Big((\Delta\mathbf{X}\mathbf{X}^\top\mathbf{L}) \odot \mathbf{M_U}\Big)\mathbf{L}^\top , \mathbf{P}2 \leftarrow \Big(((\mathbf{H} + \Delta\mathbf{X}\mathbf{X}^\top)\mathbf{L}) \odot \mathbf{M_U}\Big)\mathbf{L}^\top$
4: $\mathbf{Q} \leftarrow \mathbf{0}_{m \times n}, \mathbf{E} \leftarrow \mathbf{0}_{m \times B}$
5: **for** $i = 0, B, 2B, \dots$ **do**
6:     **for** $j = i, i+1, \dots, i+B-1$ **do**
7:         $\mathbf{Q}_{:,j} \leftarrow \text{quant}(\mathbf{W}_{:,j}^{(j)})$
8:         $\mathbf{E}_{:,j-i} \leftarrow (\mathbf{W}_{:,j}^{(j)} - \mathbf{Q}_{:,j})/\mathbf{L}_{jj}$
        $\mathbf{W}_{:,j:(i+B)} \leftarrow \mathbf{W}_{:,j:(i+B)} - \mathbf{E}_{:,j-i}\mathbf{L}_{j,j:(i+B)}^\top$
9:
        $+ \mathbf{W}_{:,j}\mathbf{P}1_{j,j:(i+B)} + (\mathbf{W}_{:,j}^{(0)} - \mathbf{W}_{:,j}^{(j)})\mathbf{P}2_{j,j:(i+B)}$
10:     **end for**
    $\mathbf{W}_{:,(i+B):} \leftarrow \mathbf{W}_{:,(i+B):} - \mathbf{E} \cdot \mathbf{L}_{i:(i+B),(i+B):}^\top$
11:
        $+ \mathbf{W}_{:,i:(i+B)}\mathbf{P}1_{i:(i+B),(i+B):} + (\mathbf{W}_{:,i:(i+B)}^{(0)} - \mathbf{W}_{:,i:(i+B)})\mathbf{P}2_{i:(i+B),(i+B):}$
12: **end for**

---

Hence, we can quantize the q-th column while only focusing on its associated residual error. With neuron decomposition, the objective in Eqn. 12 becomes:

$$\min_{\Delta\mathbf{W}_{:,q:}} ||\Delta\mathbf{W}_{:,q:}\mathbf{X}_{q:,:} - \mathbf{W}_{:,q}^{(q)}\Delta\mathbf{X}_{q,:} - (\mathbf{W}_{:,q}^{(0)} - \mathbf{W}_{:,q}^{(q)})\widetilde{\mathbf{X}}_{q,:}||_F^2$$
$$\text{s.t. } \Delta\mathbf{W}_{:,q}\mathbf{e}_q^\top + \mathbf{W}_{:,q}^{(q)} - \widehat{\mathbf{W}}_{:,q}^{(q)} = \mathbf{0}, \tag{18}$$

And the corresponding weight update becomes:

$$\Delta\mathbf{W}_{:,q:} = \frac{(\widehat{\mathbf{W}}_{:,q}^{(q)} - \mathbf{W}_{:,q}^{(q)})}{\tilde{\mathbf{H}}_{qq}^{-1}} \cdot (\tilde{\mathbf{H}}_{q,:}^{-1}) + \mathbf{W}_{:,q}^{(q)}\Delta\mathbf{X}_{q,:}\mathbf{X}_{:,q:}^\top\tilde{\mathbf{H}}_{-q}^{-1} + (\mathbf{W}_{:,q}^{(0)} - \mathbf{W}_{:,q}^{(q)})\widetilde{\mathbf{X}}_{q,:}\mathbf{X}_{:,q:}^\top\tilde{\mathbf{H}}_{-q}^{-1}. \tag{19}$$

Following GPTAQ, we can formulate $\mathbf{P}1_{q,:} = \Delta\mathbf{X}_{q,:}\mathbf{X}_{:,q:}^\top\tilde{\mathbf{H}}_{-q}^{-1}$ and $\mathbf{P}2_{q,:} = \widetilde{\mathbf{X}}_{q,:}\mathbf{X}_{:,q:}^\top\tilde{\mathbf{H}}_{-q}^{-1}$ for efficient computation. The matrix $\mathbf{P}1$ and $\mathbf{P}2$ can be precomputed in one line.

$$\mathbf{P}1 = \Big((\Delta\mathbf{X}\mathbf{X}^\top\mathbf{L}) \odot \mathbf{M_U}\Big)\mathbf{L}^\top, \mathbf{P}2 = \Big((\widetilde{\mathbf{X}}\mathbf{X}^\top\mathbf{L}) \odot \mathbf{M_U}\Big)\mathbf{L}^\top \tag{20}$$

Proof is provided in GPTAQ paper as well as Appendix A.3. We even don't need to explictly store $\widetilde{\mathbf{X}}\mathbf{X}^\top$ since $\widetilde{\mathbf{X}}\mathbf{X}^\top = \mathbf{X}\mathbf{X}^\top + \Delta\mathbf{X}\mathbf{X}^\top$. Therefore, Eqn. 19 can be written as

$$\Delta\mathbf{W}_{:,q:} = \frac{(\widehat{\mathbf{W}}_{:,q}^{(q)} - \mathbf{W}_{:,q}^{(q)})}{\tilde{\mathbf{H}}_{qq}^{-1}} \cdot (\tilde{\mathbf{H}}_{q,:}^{-1}) + \mathbf{W}_{:,q}^{(q)}\mathbf{P}1_{q,q:} + (\mathbf{W}_{:,q}^{(0)} - \mathbf{W}_{:,q}^{(q)})\mathbf{P}2_{q,q:} \tag{21}$$

Just like GPTQ and GPTAQ, the quantization results for column $q$ are only affected by updates performed on this column, and updates to later columns are irrelevant at this point. Therefore, we can *lazily update* all terms in Eqn. 21 for better GPU utilization. The full algorithm, GPTAQ combined with compensation-aware error, is summarized in Algorithm 1. Our extensions are marked in orange color.

## 5 EXPERIMENTS

**Models & Datasets.** We conduct experiments on the Llama 2 (Touvron et al., 2023) and Llama 3 (Meta, 2024) families of models, with scales ranging from 1B to 70B parameters. All models are initialized from publicly available checkpoints obtained from the Hugging Face Hub (Wolf, 2019). We evaluate performance by reporting perplexity on the WikiText-2 (Merity et al., 2016) and C4 (Raffel et al., 2020) datasets. Additionally, we assess zero-shot performance on six downstream

Table 1: Performance of 3-bit per-group symmetric weight-only quantization. We report perplexity on Wikitext2 and C4, alongside zero-shot accuracy on six downstream tasks. All models are calibrated on 128 samples from the C4 dataset following GPTAQ.

| Model | Method | Wiki2($\downarrow$) | C4($\downarrow$) | PiQA | Arc E | Arc C | HS | WG | BoolQ | Avg($\uparrow$) |
|---|---|---|---|---|---|---|---|---|---|---|
| L2-7B | FP16 | 5.47 | 7.26 | 79.0 | 74.5 | 46.3 | 76.0 | 69.0 | 77.7 | 70.4 |
| | AWQ | 6.79 | 8.93 | 77.1 | 69.9 | **41.8** | 71.2 | 67.7 | 71.5 | 66.3 |
| | GPTQ | 6.73 | 13.60 | **77.3** | 66.5 | 39.6 | 67.8 | **68.9** | 69.3 | 64.9 |
| | GPTQ+Ours | **6.40** | **8.34** | 76.8 | **70.5** | 41.2 | 71.5 | 68.1 | **71.0** | 66.5 |
| | GPTAQ | 6.53 | 8.40 | **77.7** | 67.4 | 41.3 | 72.1 | 67.6 | 71.8 | 66.3 |
| | GPTAQ+Ours | **6.25** | **8.19** | 77.4 | 69.5 | 41.5 | 72.3 | 67.4 | 71.5 | **66.6** |
| L2-13B | FP16 | 4.88 | 6.73 | 80.5 | 77.5 | 49.2 | 79.4 | 72.3 | 80.6 | 73.3 |
| | AWQ | 5.53 | 7.57 | 78.6 | 74.5 | 46.8 | 76.2 | 72.4 | 76.5 | 70.9 |
| | GPTQ | 5.43 | 7.38 | **79.2** | 75.4 | 47.2 | 75.9 | 71.0 | 79.6 | 71.4 |
| | GPTQ+Ours | **5.42** | **7.37** | 79.1 | **76.4** | **48.1** | 76.4 | 72.5 | 81.2 | 72.3 |
| | GPTAQ | 5.42 | 7.34 | **79.8** | 75.8 | 47.1 | 76.0 | 71.2 | **81.1** | 71.9 |
| | GPTAQ+Ours | **5.39** | **7.34** | 79.1 | **76.6** | 48.4 | 76.0 | 71.8 | 80.9 | **72.1** |
| L3.2-1B -Instruct | FP16 | 13.16 | 21.30 | 74.1 | 63.1 | 38.0 | 60.8 | 59.4 | 69.4 | 60.8 |
| | AWQ | 36.90 | 52.66 | 66.8 | 51.2 | 29.8 | 48.5 | 53.8 | 57.0 | 51.2 |
| | GPTQ | 21.01 | 29.14 | 70.0 | 55.9 | 32.9 | 53.3 | **57.1** | 62.9 | 55.4 |
| | GPTQ+Ours | **19.61** | **28.87** | 70.6 | 56.4 | 33.3 | 54.3 | 56.4 | **64.6** | 55.9 |
| | GPTAQ | 19.62 | 27.44 | **69.4** | 56.6 | 32.8 | 53.1 | 56.7 | 63.5 | 55.3 |
| | GPTAQ+Ours | **18.32** | **26.87** | 69.3 | 55.6 | **34.0** | 55.4 | 57.9 | 64.5 | **56.1** |
| L3-8B | FP16 | 6.14 | 9.45 | 80.9 | 77.7 | 53.2 | 79.2 | 72.9 | 81.2 | 74.2 |
| | AWQ | 9.53 | 14.74 | 76.1 | 69.2 | 42.2 | 71.4 | 69.0 | 78.2 | 67.7 |
| | GPTQ | 8.53 | 13.28 | **77.6** | 70.8 | **45.9** | 73.3 | 71.7 | 76.5 | **69.3** |
| | GPTQ+Ours | **8.00** | **12.53** | 77.4 | 68.4 | 43.4 | **74.1** | 71.9 | 78.8 | 69.0 |
| | GPTAQ | 8.39 | 12.96 | 76.9 | 72.1 | 45.0 | 70.1 | 71.0 | 77.9 | 68.8 |
| | GPTAQ+Ours | **7.77** | **12.25** | **77.7** | **73.8** | 45.7 | 74.6 | 72.3 | 79.1 | **70.5** |
| L3.1-8B -Instruct | FP16 | 7.21 | 11.39 | 80.9 | 79.6 | 54.8 | 79.1 | 74.1 | 83.9 | 75.4 |
| | AWQ | 10.50 | 16.62 | 77.3 | 65.1 | 44.7 | 72.5 | 69.5 | 80.7 | 68.3 |
| | GPTQ | 9.06 | 14.15 | 76.0 | 71.3 | 45.4 | 74.4 | 71.7 | 83.0 | 70.3 |
| | GPTQ+Ours | **8.96** | **13.97** | 78.0 | 76.3 | 50.3 | 74.7 | 73.4 | 83.5 | **72.7** |
| | GPTAQ | 8.90 | 13.85 | 76.9 | 70.5 | 46.0 | 74.5 | 71.7 | **81.5** | 70.3 |
| | GPTAQ+Ours | **8.67** | **13.79** | 78.6 | 76.2 | 50.0 | 75.1 | 72.7 | 81.3 | **72.3** |
| L3-70B | FP16 | 2.85 | 7.17 | 84.4 | 86.0 | 64.3 | 85.0 | 80.8 | 85.4 | 81.0 |
| | AWQ | 5.36 | 9.13 | **82.4** | 79.2 | 57.0 | 81.3 | 77.8 | **84.5** | 77.0 |
| | GPTQ | 5.16 | 9.23 | 81.9 | 81.5 | **57.9** | 81.7 | 78.0 | 83.9 | 77.5 |
| | GPTQ+Ours | **5.03** | **8.95** | 82.3 | 82.2 | 57.3 | 82.6 | 78.4 | 83.5 | **77.7** |
| | GPTAQ | 6.58 | 10.73 | 79.8 | 79.0 | 53.8 | **79.0** | 74.0 | **82.5** | **74.7** |
| | GPTAQ+Ours | **6.32** | **10.69** | 81.7 | 80.4 | 54.3 | 77.3 | **74.1** | 77.9 | 74.4 |

tasks: PiQA (Bisk et al., 2020), ARC Easy/Challenge (Clark et al., 2018), HellaSwag (Zellers et al., 2019), Winogrande (Sakaguchi et al., 2021), and BoolQ (Clark et al., 2019).

**Setup.** Our method is implemented in PyTorch (Paszke et al., 2019). We use per-group symmetric quantization (g128) for weights and per-token asymmetric quantization for activations. Our calibration process follows the setup of GPTAQ (Li et al., 2025).The clipping ratio for input activations is set to 0.9 as suggested in (Ashkboos et al., 2024), while the weight clipping range is searched by minimizing the mean squared error (Frantar et al., 2022). The calibration set consists of 128 samples of 2048 tokens from Wikitext2 or C4, specified in each quantization setting. Quantization for the 70B models is performed on a single NVIDIA H20 GPU with 96GB of memory, whereas all other models are quantized on a single NVIDIA A6000 GPU with 48GB of memory.

Table 2: Performance of 2-bit per-group symmetric weight-only quantization with rotation (QuaRot (Ashkboos et al., 2024)). We report perplexity on Wikitext2 and C4, alongside average zero-shot accuracy. All models are calibrated on the Wikitext2 dataset following GPTAQ.

| Precision | Method | L3.1-8B-Inst | | | L3-70B | | | L2-7B | | | L2-13B | | |
|-----------|--------|------|------|------|------|------|------|------|------|------|------|------|------|
| | | Wiki2 | C4 | Acc | Wiki2 | C4 | Acc | Wiki2 | C4 | Acc | Wiki2 | C4 | Acc |
| W2A16 | FP16 | 7.21 | 13.01 | 75.4 | 2.85 | 7.17 | 81.0 | 5.47 | 7.26 | 70.4 | 4.88 | 6.73 | 73.2 |
| | QuaRot+GPTQ | 19.8 | 53.6 | 50.7 | 30.9 | 62.6 | 45.2 | 19.0 | 36.4 | 45.0 | 10.8 | 21.9 | 50.5 |
| | QuaRot+GPTAQ | 13.9 | **33.6** | 54.7 | 11.0 | 32.8 | 58.3 | 9.5 | 19.5 | 51.5 | 7.5 | 13.9 | 55.8 |
| | QuaRot+GPTAQ+Ours | **13.6** | 34.2 | **55.9** | **10.5** | **32.1** | **59.0** | **8.9** | **18.3** | **54.0** | **7.3** | **13.6** | **58.2** |

## 5.1 WEIGHT-ONLY QUANTIZATION

We first evaluate our method in the weight-only quantization setting, a standard benchmark established by methods like GPTQ. Following established practices in GPTQ and GPTAQ, we enable act_order, which improves performance by sorting weight columns based on the Hessian diagonal magnitude. We additionally compare our method against another strong PTQ method, AWQ (Lin et al., 2023). Table 1 presents the detailed results for 3-bit quantization. By integrating our Compensation-aware Error (CAE) term into GPTQ and GPTAQ, we observe significant and consistent improvements in both perplexity and zero-shot task accuracy. For instance, integrating CAE with GPTQ on the Llama2-7B model drastically reduces C4 perplexity from 13.60 to 8.34, while increasing the average downstream accuracy from 64.9% to 66.5%. Similarly, when applied to GPTAQ, our method increases the average accuracy of the Llama3.1-8B-Instruct model from 70.3% to 72.3%. These substantial gains, observed consistently across diverse model families and scales, underscore the broad effectiveness and applicability of our proposed method.

To assess the robustness of our method under extreme compression, we extend our evaluation to the more challenging 2-bit quantization scenario. To mitigate the inherent difficulty of this setting, we incorporate QuaRot (Ashkboos et al., 2024), a training-free weight rotation technique. As shown in Table 2, our method yields significant performance improvements even when integrated with baselines already enhanced by the QuaRot technique. For example, on the Llama2-13B model, our approach further reduces Wikitext2 perplexity from 7.50 to 7.32 and improves the average accuracy from 55.8% to 58.2% over the QuaRot+GPTAQ baseline. These results provide strong evidence that our proposed error term more accurately models and compensates for the complex errors introduced by low-bit quantization, thereby recovering model performance under these stringent conditions.

## 5.2 WEIGHT-ACTIVATION QUANTIZATION

To further validate the efficacy of our method, we extend our evaluation to the challenging weight-and-activation quantization setting. To address the significant performance degradation caused by activation outliers, we incorporate two rotation-based transformations: the tuning-free QuaRot (Ashkboos et al., 2024) and the optimized SpinQuant (Liu et al., 2024). For SpinQuant, we directly utilize the official pre-trained rotation matrices without additional fine-tuning. Given GP-TAQ's superior performance when activation is quantized, we primarily evaluate the performance of GPTAQ+Ours. The results for GPTQ+Ours are deferred to Appendix A.4. As presented in Table 3, integrating our method with these advanced baselines yields superior performance across most evaluations in the stringent W2A4KV4 scenario. On the Llama2-13B model, our approach achieves a significant advantage, reducing Wikitext2 perplexity from 9.55 (SpinQuant+GPTAQ) to 8.60, while increasing the average downstream task accuracy from 50.2% to 52.2%. Similar performance gains are observed on the Llama2-7B model, where perplexity decreases from 11.6 to 11.1 and average accuracy increases from 48.1% to 49.2%. For the Llama3-8B model, our method achieves 1.3% higher average accuracy compared to Quarot+GPTAQ.

A notable finding emerges from our experiments on Llama3-70B. We observe that while QuaRot-based methods remain stable, SpinQuant-based approaches suffer from catastrophic performance degradation (Perplexity > 1e5). We hypothesize that this failure stems from the pre-trained rotation matrix. As it is optimized under a W16A4KV4 setting, it cannot adapt to the substantial distribution shifts induced by 2-bit weight quantization. Overall, these results indicate that our pro-

Table 3: Performance of W2A4KV4 quantization. We report perplexity on Wikitext2 and C4, alongside zero-shot accuracy on six downstream tasks. All models are calibrated on 128 samples from the Wikitext2 dataset following GPTAQ.

| Model | Method | Wiki2($\downarrow$) | C4($\downarrow$) | PiQA | Arc E | Arc C | HS | WG | BoolQ | Avg($\uparrow$) |
|---|---|---|---|---|---|---|---|---|---|---|
| L2-7B | FP16 | 5.47 | 7.26 | 79.0 | 74.5 | 46.3 | 76.0 | 69.0 | 77.7 | 70.4 |
| | SpinQ+GPTQ | 31.9 | 61.3 | 57.1 | 34.9 | 23.6 | 33.2 | 53.7 | 61.4 | 44.0 |
| | SpinQ+GPTAQ | 11.6 | 26.5 | 62.2 | 42.3 | 25.5 | 40.9 | 54.7 | **63.4** | 48.1 |
| | SpinQ+GPTAQ+Ours | **11.1** | **24.7** | 62.7 | 45.4 | 27.6 | 42.3 | 55.0 | 62.3 | **49.2** |
| | Quarot+GPTQ | 30.0 | 48.9 | 55.7 | 35.2 | 23.9 | 32.4 | 50.6 | 61.8 | 43.4 |
| | Quarot+GPTAQ | 11.7 | 24.8 | 62.3 | 45.6 | **25.6** | **41.2** | 54.0 | **62.4** | 48.5 |
| | Quarot+GPTAQ+Ours | **11.5** | **23.6** | 63.6 | 46.4 | 24.9 | 40.8 | 56.0 | 61.9 | **48.9** |
| L2-13B | FP16 | 4.88 | 6.73 | 80.5 | 77.5 | 49.2 | 79.4 | 72.3 | 80.6 | 73.3 |
| | DuQuant+LWC | 16.4 | - | 58.7 | 37.3 | 24.9 | 41.5 | 53.3 | 62.0 | 46.2 |
| | SpinQ+GPTQ | 13.3 | 33.6 | 59.0 | 39.5 | 24.9 | 40.3 | 52.6 | 61.4 | 46.3 |
| | SpinQ+GPTAQ | 9.55 | 62.1 | 62.3 | 44.6 | 27.8 | 48.0 | 55.0 | 63.6 | 50.2 |
| | SpinQ+GPTAQ+Ours | **8.60** | **20.3** | 63.4 | 48.7 | 29.1 | 51.2 | 56.8 | 64.1 | **52.2** |
| | Quarot+GPTQ | 12.5 | 26.1 | 61.6 | 45.7 | 27.0 | 44.0 | 55.8 | 62.3 | 49.4 |
| | Quarot+GPTAQ | 8.89 | 17.2 | 65.3 | **47.2** | 27.3 | 48.5 | 57.7 | 63.0 | 51.5 |
| | Quarot+GPTAQ+Ours | **8.61** | **16.5** | 66.5 | 44.0 | 30.1 | 48.9 | 58.9 | 63.2 | 51.9 |
| L3-8B | FP16 | 6.14 | 9.45 | 80.9 | 77.7 | 53.2 | 79.2 | 72.9 | 81.2 | 74.2 |
| | DuQuant+LWC | 4e4 | - | 51.3 | 26.1 | 25.6 | 25.5 | 49.3 | 37.9 | 35.6 |
| | SpinQ+GPTQ | 46.9 | 163 | 51.5 | 25.5 | 25.6 | 33.1 | 51.7 | 57.6 | 40.8 |
| | SpinQ+GPTAQ | 18.3 | 55.6 | **61.4** | **42.3** | 26.6 | 40.1 | **54.7** | **62.7** | 47.9 |
| | SpinQ+GPTAQ+Ours | **18.1** | **53.7** | 60.6 | 41.4 | **28.1** | 40.8 | 54.1 | 62.1 | 47.9 |
| | Quarot+GPTQ | 45.2 | 88.7 | 55.4 | 34.7 | 20.7 | 31.4 | 49.7 | 44.3 | 39.4 |
| | Quarot+GPTAQ | 23.0 | 62.8 | 55.8 | 37.2 | 22.3 | **36.5** | 50.4 | 59.3 | 43.6 |
| | Quarot+GPTAQ+Ours | **22.1** | **61.5** | 56.0 | 38.5 | 24.6 | 35.8 | 53.5 | 60.9 | **44.9** |
| L3-70B | FP16 | 2.85 | 7.17 | 84.4 | 86.0 | 64.3 | 85.0 | 80.8 | 85.4 | 81.0 |
| | SpinQ+GPTQ | 6378 | > 1e5 | **53.7** | **31.0** | 23.8 | 27.8 | 51.3 | 46.7 | 39.0 |
| | SpinQ+GPTAQ | 5004 | > 1e5 | 52.7 | 27.4 | 26.0 | 29.9 | 52.4 | 49.6 | 39.6 |
| | SpinQ+GPTAQ+Ours | **3282** | **5e4** | 52.2 | 26.1 | **26.4** | 30.2 | 52.8 | 50.8 | **39.9** |
| | Quarot+GPTQ | 56.2 | 91.6 | 54.3 | 30.8 | 19.5 | 30.3 | 50.8 | 56.6 | 40.4 |
| | Quarot+GPTAQ | 30.1 | 57.0 | **57.5** | 35.1 | **23.1** | 31.9 | **51.8** | 60.8 | 43.3 |
| | Quarot+GPTAQ+Ours | **26.5** | **49.5** | 57.4 | 37.0 | 22.2 | 32.7 | 50.3 | 61.1 | **43.5** |

Table 4: Ablation study of $\Delta \mathbf{W}$s. We report perplexity and average zero-shot Accuracy.

| Precision | Method | $\Delta \mathbf{W}$ | L2-7B | | | L2-13B | | |
|---|---|---|---|---|---|---|---|---|
| | | | Wiki2 | C4 | Acc | Wiki2 | C4 | Acc |
| W2A16 | GPTQ | $\mathbf{E}_{:,q}\mathbf{L}_{q,q:}^T$ | 19.0 | 36.5 | 44.9 | 10.8 | 21.9 | 50.5 |
| | GPTQ+Ours | $\mathbf{E}_{:,q}\mathbf{L}_{q,q:}^T + (\mathbf{W}_{:,q}^{(0)} - \mathbf{W}_{:,q}^{(q)})\mathbf{P}2_{q,q:}$ | **17.9** | **35.4** | **47.5** | **9.7** | **19.8** | **54.3** |
| | GPTAQ | $\mathbf{E}_{:,q}\mathbf{L}_{q,q:}^T + \mathbf{W}_{:,q}^{(q)}\mathbf{P}1_{q,q:}$ | 9.5 | 19.5 | 51.5 | 7.5 | 13.9 | 55.8 |
| | GPTAQ+Ours | $\mathbf{E}_{:,q}\mathbf{L}_{q,q:}^T + \mathbf{W}_{:,q}^{(q)}\mathbf{P}1_{q,q:} + (\mathbf{W}_{:,q}^{(0)} - \mathbf{W}_{:,q}^{(q)})\mathbf{P}2_{q,q:}$ | **8.9** | **18.3** | **54.0** | **7.3** | **13.6** | **58.2** |

posed compensation-aware error remains effective at addressing the complex errors introduced by simultaneous weight and activation quantization, robustly lowering model perplexity and improving downstream task accuracy.

## 5.3 ABLATION STUDY

Our proposed weight update, $\Delta \mathbf{W}$, introduces a new term, $(\mathbf{W}_{:,q}^{(0)} - \mathbf{W}_{:,q}^{(q)})\mathbf{P}2_{q,q:}$, which is designed to account for the discrepancy between the pre-quantization compensated weights and the original weights. To isolate and validate the contribution of our proposed term, we conduct an ablation study by integrating it into two strong baseline frameworks: GPTQ and GPTAQ. While Table 1 already serves as a comprehensive ablation study, here we additionally present the results for extremely low-bit quantization combined with rotation. As presented in Table 4, our term delivers consistent and substantial performance gains when integrated into either framework. When added to GPTQ, our term improves the average accuracy on Llama2-7B from 44.9% to 47.5% and on Llama2-13B from

Table 5: GPU Memory (GB) needed to **perform calibration** (not quantized inference) on LLaMA2-7B and LLaMA3-70B following the standard setup. 'Peak' denotes the peak memory usage during the whole calibration process, obtained by the nvitop command. † denotes moving the fp activation cache to cpu.

| Llama2-7B | q_proj | k_proj | v_proj | o_proj | up_proj | gate_proj | down_proj | Peak |
|---|---|---|---|---|---|---|---|---|
| $m, n$ | 4096, 4096 | 4096, 4096 | 4096, 4096 | 4096, 4096 | 11008, 4096 | 11008, 4096 | 4096, 11008 | |
| GPTQ | 0.13GB | 0.13GB | 0.13GB | 0.13GB | 0.29GB | 0.29GB | 0.48GB | 8.5GB |
| GPTAQ | 0.16GB | 0.16GB | 0.16GB | 0.16GB | 0.32GB | 0.32GB | 0.70GB | 19.8/10.1$^\dagger$GB |
| GPTAQ+Ours | 0.25GB | 0.25GB | 0.25GB | 0.25GB | 0.52GB | 0.52GB | 1.11GB | 20.6/11.0$^\dagger$GB |
| **Llama3-70B** | q_proj | k_proj | v_proj | o_proj | up_proj | gate_proj | down_proj | Peak |
| $m, n$ | 8192, 8192 | 8192, 8192 | 8192, 8192 | 8192, 8192 | 28672, 8192 | 28672, 8192 | 8192, 28672 | |
| GPTQ | 0.52GB | 0.52GB | 0.52GB | 0.52GB | 1.49GB | 1.49GB | 2.92GB | 23.2GB |
| GPTAQ | 0.65GB | 0.65GB | 0.65GB | 0.65GB | 1.62GB | 1.62GB | 4.48GB | 63.7/35.8$^\dagger$GB |
| GPTAQ+Ours | 1.03GB | 1.03GB | 1.03GB | 1.03GB | 2.64GB | 2.64GB | 6.95GB | 69.5/41.5$^\dagger$GB |

Table 6: Quantization Time (s) of 3bit weight-only quantization on a single NVIDIA H20 GPU. We report the mean and standard deviation over 4 measurements.

| Model | GPTQ | GPTAQ | GPTAQ+Ours |
|---|---|---|---|
| Llama2-7b | 796±10 | 952±30 | 1001±12 |
| Llama2-13B | 1354±3 | 1626±11 | 1728±17 |
| Llama3-70B | 4746±120 | 5883±123 | 6344±153 |

50.5% to 54.3%, with corresponding perplexity reductions. This demonstrates that accounting for the compensation error is beneficial even without cross-layer error propagation. More importantly, when combined with GPTAQ, the performance is further enhanced. On Llama2-13B, this combination lowers the Wikitext2 perplexity to 7.3 and increases the average accuracy to 58.2%. This result confirms that our compensation-aware error is complementary to existing residual error terms and that explicitly modeling the error from the weight update process.

### 5.4 ALGORITHM EFFICIENCY

First, regarding calibration memory, the matrix needed for calibration are summarized in Table 10. Our approach (Algorithm 1) mainly introduces additional storage requirements over GPTAQ for two components: $\mathbf{W}^{(0)} \in \mathbb{R}^{m \times n}$ and $\mathbf{P}2 \in \mathbb{R}^{n \times n}$. While this increases the peak memory usage, this overhead is manageable and strictly confined to the offline quantization process. As shown in Table 5, the per-layer memory footprint remains practical for modern hardware. **Crucially, the memory footprint of the final quantized model at inference is unaffected.**

Second, in terms of quantization time, our method incurs a minimal overhead of only ∼5% compared to GPTAQ for the end-to-end model quantization process (Table 6). This marginal cost highlights the efficiency of the neuron decomposition technique, which seamlessly integrates the compensation-aware error into the weight update computation with negligible impact on quantization time. **Notably, our method doesn't incur run-time overhead when inferencing with quantized weights.**

## 6 CONCLUSION

In this work, we analyze and refine the calibration objective in compensation-based quantization methods such as GPTQ and GPTAQ. We establish that the residual error should originate not only from the preceding layer's output error but also from the discrepancy introduced by the weight compensation process itself. Based on this finding, we introduce the compensation-aware error to the residual error formulation. As demonstrated through extensive experiments, our proposed enhancements are efficiently and seamlessly integrated into existing frameworks and boost their performance. The results confirm that our method consistently and significantly boosts quantization performance across a diverse range of large language models and quantization settings, underscoring the critical impact of precise error modeling in post-training quantization.

ACKNOWLEDGEMENTS

This work was supported by the National Science and Technology Major Project (2026ZD1305800), Ningbo Key Research and Development Program (2025Z082), Zhejiang Province's Leading Talent Project in Science and Technology Innovation (2023R5204), and Zhejiang University - Vivo Information Technology Joint Research Center.

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

# A APPENDIX

## A.1 INTEGRATION OF OUR METHOD WITH GPTQ

Since our focus is on the residual error introduced by weight compensation—an error present in both GPTQ and GPTAQ—it is still sufficient to incorporate the P2 term into the weight compensation. Therefore, our method can be integrated with both GPTQ and GPTAQ. GPTQ consistently uses the input to the quantized model, $\mathbf{X}$, as its input, which means $\Delta\mathbf{X} = 0$, eliminating the need for the P1 term. The detailed process is illustrated in algorithm 2.

---

**Algorithm 2** GPTQ quantization with Compensation-aware Error

1: **Input:** FP weight $\mathbf{W}$, calibration input $\mathbf{X}$, and Block size $B$
2: $\mathbf{W}^{(0)} \leftarrow \mathbf{W}, \mathbf{H} \leftarrow \mathbf{X}\mathbf{X}^\top, \Delta\mathbf{X}\mathbf{X}^\top \leftarrow (\tilde{\mathbf{X}} - \mathbf{X})\mathbf{X}^\top, \mathbf{L} = Inverse\_Cholesky(\mathbf{H} + \lambda_1\mathbf{I})$
3: $\mathbf{P2} \leftarrow \left( ((\mathbf{H} + \Delta\mathbf{X}\mathbf{X}^\top)\mathbf{L}) \odot \mathbf{M_U} \right)\mathbf{L}^\top$
4: $\mathbf{Q} \leftarrow \mathbf{0}_{m \times n}, \mathbf{E} \leftarrow \mathbf{0}_{m \times B}$
5: **for** $i = 0, B, 2B, \ldots$ **do**
6:     **for** $j = i, i+1, \ldots, i+B-1$ **do**
7:         $\mathbf{Q}_{:,j} \leftarrow \text{quant}(\mathbf{W}_{:,j}^{(j)})$
8:         $\mathbf{E}_{:,j-i} \leftarrow (\mathbf{W}_{:,j}^{(j)} - \mathbf{Q}_{:,j})/\mathbf{L}_{jj}$
9:         $\mathbf{W}_{:,j:(i+B)}^{(j+1)} \leftarrow \mathbf{W}_{:,j:(i+B)}^{(j)} - \mathbf{E}_{:,j-i}\mathbf{L}_{j,j:(i+B)}^\top + (\mathbf{W}_{:,j}^{(0)} - \mathbf{W}_{:,j}^{(j)})\mathbf{P2}_{j,j:(i+B)}$
10:     **end for**
11:     $\mathbf{W}_{:,(i+B):} \leftarrow \mathbf{W}_{:,(i+B):} - \mathbf{E} \cdot \mathbf{L}_{i:(i+B),(i+B):}^\top + (\mathbf{W}_{:,i:(i+B)}^{(0)} - \mathbf{W}_{:,i:(i+B)})\mathbf{P2}_{i:(i+B),(i+B):}$
12: **end for**

---

## A.2 ADDITIONAL RESULTS ON VISION TRANSFORMER

To evaluate the generalizability of our method beyond language models, we conduct experiments on Vision Transformers. We use DeiT-Tiny, DeiT-Small, and DeiT-Base models (Touvron et al., 2021) and select 128 samples from the ImageNet (Deng et al., 2009) training dataset for calibration data. The compared baselines include recent post-training quantization methods such as APQ-ViT (Ding et al., 2022), RepQ-ViT (Li et al., 2023), GPTQ (Frantar et al., 2022), and GPTAQ (Li et al., 2025). For all compensation-based methods (GPTQ, GPTAQ, and Ours), we utilize the `act_order` option to sort weight columns by Hessian magnitude, as this proves beneficial for performance. We test under both W4A4 and the more challenging W2A4 quantization settings.

The results are presented in Table 7. Under W4A4 quantization, our method achieves highly competitive performance across all models. On DeiT-Small, our approach reaches 74.0% accuracy, outperforming the strong GPTAQ baseline. The advantages of our method become more pronounced in the lower-precision W2A4 setting. For DeiT-Base, our approach improves the accuracy to 62.1%, a notable gain over both GPTQ (51.8%) and GPTAQ (61.3%). These results on vision transformers confirm that our proposed compensation-aware error formulation is effective and robust, providing consistent performance improvements even in highly compressed, non-linguistic models.

Table 7: Quantization results of Vision Transformers on ImageNet (accuracy ↑). We evaluate three DeiT models under W4A4 and W2A4 quantization.

| Model | FP16 | W4A4 | | | | | W2A4 | | |
|---|---|---|---|---|---|---|---|---|---|
| | - | APQ-ViT | RepQ-ViT | GPTQ | GPTAQ | Ours | GPTQ | GPTAQ | Ours |
| DeiT-Tiny | 72.0 | - | - | 64.5 | 65.6 | **65.7** | 27.5 | 26.8 | **27.8** |
| DeiT-Small | 79.8 | 34.1 | 71.8 | 69.0 | 73.8 | **74.0** | 40.4 | 45.7 | **46.5** |
| DeiT-Base | 81.3 | 64.4 | 75.6 | 77.6 | 78.1 | 78.0 | 51.8 | 61.3 | **62.1** |

Table 8: Additional results of W2A4KV4 quantization. We report perplexity on Wikitext2 and C4, alongside zero-shot accuracy on six downstream tasks. All models are calibrated on 128 samples from the Wikitext2 dataset.

| Model | Method | Wiki2($\downarrow$) | C4($\downarrow$) | PiQA | Arc E | Arc C | HS | WG | BoolQ | Avg($\uparrow$) |
|-------|--------|-------|------|------|-------|-------|------|------|-------|------|
| | FP16 | 5.47 | 7.26 | 79.0 | 74.5 | 46.3 | 76.0 | 69.0 | 77.7 | 70.4 |
| L2-7B | SpinQ+GPTQ | 31.9 | 61.3 | 57.1 | 34.9 | 23.6 | 33.2 | **53.7** | **61.4** | **44.0** |
| | SpinQ+GPTQ+Ours | **23.8** | **44.3** | **57.7** | **35.5** | 23.6 | **35.0** | 53.0 | 56.6 | 43.8 |
| | FP16 | 5.47 | 7.26 | 79.0 | 74.5 | 46.3 | 76.0 | 69.0 | 77.7 | 70.4 |
| L3-8B | SpinQ+GPTQ | 46.9 | 163 | 51.5 | 25.5 | **25.6** | 33.1 | 51.7 | 57.6 | 40.8 |
| | SpinQ+GPTQ+Ours | **24.3** | **80.0** | **57.3** | **39.5** | 24.4 | **35.7** | **52.2** | **57.8** | **44.4** |
| | FP16 | 4.88 | 6.73 | 80.5 | 77.5 | 49.2 | 79.4 | 72.3 | 80.6 | 73.3 |
| L2-13B | SpinQ+GPTQ | 13.3 | 33.6 | 59.0 | 39.5 | 24.9 | 40.3 | 52.6 | 61.4 | 46.3 |
| | SpinQ+GPTQ+Ours | **11.7** | **25.4** | **62.8** | **45.5** | **26.4** | **44.1** | **55.8** | **62.3** | **49.5** |

### A.3 PROOF OF PRECOMPUTATION $\mathbf{P}$

A complete proof is provided in the GPTAQ paper. Here, we briefly recapitulate it using the computation of $\mathbf{P}1$ as an example. For any further questions, we refer the reader to the proof in Appendix A.3 of the GPTAQ paper.

*Proof.* We derive P1 row-wise. Starting from the expression

$$\mathbf{P}1_{i,:} = \Delta\mathbf{X}_{i,:}\mathbf{X}^\top\mathbf{L}_{i+1:,i+1:}\mathbf{L}^\top_{i+1:,i+1:},$$

note that $\mathbf{L}^\top_{i+1:,i+1:}$ has support only in columns $j > i$. Thus, $\mathrm{P}1_{i,j} = 0$ for $i \geq j$, and for $i < j$,

$$\mathbf{P}1_{i,j} = \sum_{a=i+1}^{j} \mathbf{O}_{i,a}\mathbf{L}^\top_{a,j},$$

where $\mathbf{O}_{i,a} := (\Delta\mathbf{X}_{i,:}\mathbf{X}^\top\mathbf{L}_{i+1:,i+1:})_a$.

Since $\mathbf{L}_{i+1:,i+1:}$ is zero in columns $a \leq i$, we have $\mathbf{O}_{i,a} = 0$ for $a \leq i$, and for $a > i$,

$$\mathbf{O}_{i,a} = \sum_{b=a}^{n} (\Delta\mathbf{X}\mathbf{X}^\top)_{i,b}\,\mathbf{L}_{b,a}.$$

This means $\mathbf{O} = (\Delta\mathbf{X}\mathbf{X}^\top\mathbf{L}) \odot \mathbf{M_U}$, where $\odot$ denotes element-wise multiplication and $\mathbf{M_U}$ masks out the lower triangle (including diagonal).

Finally, since $\mathbf{O}$ is strictly upper-triangular, multiplying by $\mathbf{L}^\top$ yields:

$$(\mathbf{O}\mathbf{L}^\top)_{i,j} = \sum_{a=1}^{n} \mathbf{O}_{i,a}\mathbf{L}^\top_{a,j} = \sum_{a=i+1}^{j} \mathbf{O}_{i,a}\mathbf{L}^\top_{a,j} = \mathbf{P}_{i,j},$$

because $\mathbf{O}_{i,a} = 0$ for $a \leq i$ and $\mathbf{L}^\top_{a,j} = 0$ for $a > j$. Hence,

$$\boxed{\mathbf{P}1 = \left((\Delta\mathbf{X}\mathbf{X}^\top\mathbf{L}) \odot \mathbf{M_U}\right)\mathbf{L}^\top}.$$

$\square$

The computational efficiency of $\mathbf{P}1$ stems from the triangular property of $\mathbf{L}$, independent of $\Delta\mathbf{X}$. Analogously, P2 can be computed as:

$$\mathbf{P}2 = \left((\tilde{\mathbf{X}}\mathbf{X}^\top\mathbf{L}) \odot \mathbf{M_U}\right)\mathbf{L}^\top$$

## A.4 Additional Results on WEIGHT-ACTIVATION QUANTIZATION

As shown in Table 8, we provide the results of combining our method with GPTQ for weight-activation quantization, as a supplement to Table 3. This further validates the generalizability of our method.

## A.5 ADDITIONAL RESULTS ON QWEN MODELS

To further illustrate the generalizability of our proposed method, we conducted experiments on the Qwen series models, with results presented in Table 9. On the 4B, 8B, and 14B models, our method achieved significant and stable improvements in both PPL (perplexity) and average accuracy compared to GPTAQ, further validating the generalizability of our approach.

Table 9: Performance of 2-bit per-group symmetric weight-only quantization on Qwen3 models. We report perplexity on C4, alongside zero-shot accuracy on six downstream tasks. All models are calibrated on 128 samples from the C4 dataset following GPTAQ.

| Model | Method | C4($\downarrow$) | Arc E | Arc C | HS | WG | BoolQ | PiQA | Avg($\uparrow$) |
|---|---|---|---|---|---|---|---|---|---|
| | FP16 | 19.85 | 78.5 | 54.0 | 68.4 | 66.1 | 85.1 | 74.8 | 71.2 |
| Qwen3-4B | GPTAQ | 57.1 | 37.6 | 23.0 | 34.6 | 53.1 | 63.7 | 61.3 | 45.6 |
| | GPTAQ+Ours | **39.9** | **47.1** | **28.4** | **45.1** | **56.8** | **70.7** | **65.3** | **52.2** |
| | FP16 | 15.4 | 80.9 | 56.7 | 74.9 | 67.8 | 86.6 | 77.8 | 74.1 |
| Qwen3-8B | GPTAQ | 32.9 | 50.5 | **29.9** | 45.4 | 56.8 | 69.2 | 65.6 | 52.9 |
| | GPTAQ+Ours | **29.9** | **54.3** | 29.7 | **48.4** | **58.5** | **71.4** | **66.5** | **54.8** |
| | FP16 | 13.82 | 82.8 | 60.2 | 78.9 | 72.8 | 89.4 | 79.9 | 77.3 |
| Qwen3-14B | GPTAQ | 21.4 | 65.0 | 39.4 | 61.0 | **66.4** | 83.4 | **74.1** | 64.9 |
| | GPTAQ+Ours | **20.9** | **65.4** | **40.1** | **61.2** | 66.1 | **83.8** | 73.6 | **65.1** |

We present the sizes (dimensions) of the matrices required for calibration in Table. 10.

Table 10: Matrix needed to perform calibration and their sizes. $C_o$ and $C_i$ denotes the output-channel and input-channel of weights, while $b$ denotes the blocksize for lazy-batch update.

| Matrix | GPTQ | GPTAQ | GPTAQ+Ours |
|---|---|---|---|
| Original weight: $W^{(0)}$ | - | - | $C_o \times C_i$ |
| Compensated weight: $W$ | $C_o \times C_i$ | $C_o \times C_i$ | $C_o \times C_i$ |
| Fake quant weight: $Q$ | $C_o \times C_i$ | $C_o \times C_i$ | $C_o \times C_i$ |
| Cholesky factor: $L$ | $C_i \times C_i$ | $C_i \times C_i$ | $C_i \times C_i$ |
| Precompute 1: $P1$ | - | $C_i \times C_i$ | $C_i \times C_i$ |
| Precompute 2: $P2$ | - | - | $C_i \times C_i$ |
| In-block weight: $W_b$ | $C_o \times b$ | $C_o \times b$ | $C_o \times b$ |
| In-block Error: $E_b$ | $C_o \times b$ | $C_o \times b$ | $C_o \times b$ |
| In-block quant weight: $Q_b$ | $C_o \times b$ | $C_o \times b$ | $C_o \times b$ |
| In-block cholesky: $L_b$ | $b \times b$ | $b \times b$ | $b \times b$ |
| In-block precompute: $P_b$ | - | $b \times b$ | $2 \times b \times b$ |

Table 11: Performance of 3-bit per-group symmetric weight-only quantization. using different numbers of calibration samples (64, 96, 128, 192). Lower C4 perplexity and higher average accuracy are better.

| Model | #Samples | Method | C4($\downarrow$) | Arc E | Arc C | HS | WG | BoolQ | PiQA | Avg($\uparrow$) |
|---|---|---|---|---|---|---|---|---|---|---|
| L2-7B | - | FP16 | 7.26 | 79.0 | 74.5 | 46.3 | 76.0 | 69.0 | 77.7 | 70.4 |
| | 64 | GPTAQ | 8.44 | **68.8** | **41.9** | 71.8 | 67.6 | 72.4 | 77.2 | 66.5 |
| | | GPTAQ+Ours | **8.31** | 68.4 | 41.1 | **72.0** | 67.6 | **72.5** | **77.3** | **66.6** |
| | 96 | GPTAQ | 8.42 | 68.1 | **41.7** | 72.2 | 67.0 | 71.9 | 77.5 | 66.4 |
| | | GPTAQ+Ours | **8.23** | **68.9** | 41.6 | 72.2 | **67.2** | **74.2** | **77.6** | **66.9** |
| | 128 | GPTAQ | 8.40 | 67.4 | 41.3 | 72.1 | **67.6** | 71.8 | **77.7** | 66.3 |
| | | GPTAQ+Ours | **8.19** | **69.5** | **41.5** | **72.3** | 67.4 | 71.5 | 77.4 | **66.6** |
| | 192 | GPTAQ | 8.38 | 67.4 | 40.7 | 71.8 | **68.4** | 73.4 | 78.1 | 66.6 |
| | | GPTAQ+Ours | **8.19** | **70.0** | **41.6** | **72.4** | 67.7 | **75.7** | **78.2** | **67.6** |
| L3.2-1B -Instruct | - | FP16 | 21.3 | 63.1 | 38.0 | 60.8 | 59.4 | 69.4 | 74.1 | 60.8 |
| | 64 | GPTAQ | 28.4 | 55.8 | 32.2 | 53.9 | **56.4** | 63.3 | 69.8 | 55.2 |
| | | GPTAQ+Ours | **27.7** | **56.9** | **33.9** | **54.7** | 56.3 | **64.4** | **70.4** | **56.1** |
| | 96 | GPTAQ | 27.4 | **57.1** | **32.6** | 53.5 | 55.6 | **64.2** | 68.9 | 55.3 |
| | | GPTAQ+Ours | **26.8** | 57.0 | 31.6 | **54.8** | **58.6** | 63.2 | **69.4** | **55.8** |
| | 128 | GPTAQ | 27.4 | **56.6** | 32.8 | 53.1 | 56.7 | 63.5 | **69.4** | 55.3 |
| | | GPTAQ+Ours | **26.9** | 55.6 | **34.0** | **55.4** | **57.9** | **64.5** | 69.3 | **56.1** |
| | 192 | GPTAQ | 27.1 | 53.2 | 29.8 | 54.4 | 56.3 | 63.0 | 67.4 | 54.0 |
| | | GPTAQ+Ours | **26.9** | **57.0** | **33.2** | **54.8** | **56.5** | **63.6** | **69.5** | **55.8** |
| L3.1-8B | - | FP16 | 9.54 | 81.1 | 53.4 | 78.9 | 73.5 | 82.1 | 81.3 | 75.1 |
| | 64 | GPTAQ | 12.59 | **75.8** | 47.8 | **74.5** | 72.4 | 78.8 | **79.3** | 71.4 |
| | | GPTAQ+Ours | **12.37** | 75.7 | **48.0** | 74.1 | **73.2** | **80.4** | 78.9 | **71.7** |
| | 96 | GPTAQ | 12.82 | 73.6 | 46.3 | 74.8 | 70.6 | 80.2 | 78.6 | 70.7 |
| | | GPTAQ+Ours | **12.21** | **75.1** | **47.8** | 74.8 | **71.4** | **80.5** | **79.0** | **71.4** |
| | 128 | GPTAQ | 12.26 | **74.4** | 47.8 | 75.0 | 72.0 | 79.4 | 77.3 | 71.0 |
| | | GPTAQ+Ours | **12.08** | 73.7 | **47.9** | **75.2** | **73.2** | **80.0** | **79.5** | **71.6** |
| | 192 | GPTAQ | 12.32 | 76.1 | 48.1 | **75.2** | 71.3 | **80.4** | 78.2 | 71.5 |
| | | GPTAQ+Ours | **12.06** | **76.8** | **49.7** | 74.8 | **71.7** | 79.5 | **78.6** | **71.8** |
| L3-8B | - | FP16 | 9.45 | 77.7 | 53.2 | 79.2 | 72.9 | 81.2 | 80.9 | 74.2 |
| | 64 | GPTAQ | 12.83 | **73.8** | 45.0 | 71.5 | 70.7 | 78.0 | 78.1 | 69.5 |
| | | GPTAQ+Ours | **12.62** | 73.2 | **46.3** | **73.0** | **71.2** | 78.0 | **78.4** | **70.0** |
| | 96 | GPTAQ | 12.70 | 72.6 | 44.8 | 74.0 | 71.7 | 79.3 | **78.8** | 70.7 |
| | | GPTAQ+Ours | **12.24** | **74.5** | **46.5** | **75.0** | **72.2** | **80.1** | 78.8 | **71.2** |
| | 128 | GPTAQ | 12.96 | 72.1 | 45.0 | 70.1 | 71.0 | 77.9 | 76.9 | 68.8 |
| | | GPTAQ+Ours | **12.25** | **73.8** | **45.7** | **74.6** | **72.3** | **79.1** | **77.7** | **70.5** |
| | 192 | GPTAQ | 12.36 | 72.3 | 44.2 | 75.2 | **72.9** | 77.1 | 77.6 | 69.9 |
| | | GPTAQ+Ours | **12.09** | **75.6** | **48.6** | **75.6** | 72.1 | **79.8** | **79.5** | **71.9** |

## A.6 ANALYSIS ON SENSITIVITY TO CALIBRATION DATASETS

We first analyzed the impact of the number of samples on calibration quality. We conducted validation experiments on four models using 64, 96, 128, and 192 samples, respectively, with a 3-bit per-group weight-only quantization setting. The results are summarized in Table. 11. Our method consistently achieves a stable improvement over GPTAQ, demonstrating its robustness across vary-

ing numbers of calibration samples. Subsequently, we analyzed the influence of the calibration set

Table 12: Performance of 3-bit per-group symmetric weight-only quantization. using different calibration datasets. Lower C4 perplexity and higher average accuracy are better. 'red_stack' denotes the stackexange subset of redpajama, and 'red_cc' denotes the commoncrawl subset of redpajama.

| Model | Dataset | Method | Wiki2($\downarrow$) | C4($\downarrow$) | Arc E | Arc C | HS | WG | BoolQ | PiQA | Avg($\uparrow$) |
|---|---|---|---|---|---|---|---|---|---|---|---|
| L2-7B | - | FP16 | 5.47 | 7.26 | 79.0 | 74.5 | 46.3 | 76.0 | 69.0 | 77.7 | 70.4 |
| | red_stack | GPTAQ | **6.54** | 8.75 | 68.9 | 39.5 | 71.0 | 66.9 | 71.0 | 76.4 | 65.6 |
| | | GPTAQ+Ours | 6.56 | **8.72** | **71.0** | **41.5** | **71.5** | **68.1** | **72.0** | **77.0** | **66.8** |
| | red_cc | GPTAQ | 6.42 | 8.56 | 68.0 | **42.0** | **71.7** | 66.5 | **74.3** | 76.6 | 66.5 |
| | | GPTAQ+Ours | **6.23** | **8.33** | **68.2** | 41.0 | 71.2 | **68.0** | 73.6 | **76.9** | **66.6** |
| | c4 | GPTAQ | 6.54 | 8.38 | **67.4** | 40.7 | 71.8 | **68.4** | 73.4 | 78.1 | 66.6 |
| | | GPTAQ+Ours | **6.25** | **8.19** | 67.0 | **41.6** | **72.4** | 67.7 | **75.7** | **78.2** | **67.6** |
| | wiki2 | GPTAQ | 5.96 | 9.22 | 69.2 | 40.7 | 71.4 | 66.0 | **71.7** | 76.8 | 66.0 |
| | | GPTAQ+Ours | **5.93** | **8.56** | **71.5** | **42.2** | 71.7 | **68.2** | 71.4 | **77.0** | **67.0** |
| L3-8B | - | FP16 | 6.14 | 9.45 | 77.7 | 53.2 | 79.2 | 72.9 | 81.2 | 80.9 | 74.2 |
| | red_stack | GPTAQ | 10.41 | 14.56 | 68.8 | 42.1 | 70.8 | 69.7 | 74.2 | 76.3 | 67.0 |
| | | GPTAQ+Ours | **9.16** | **14.22** | **68.9** | **44.1** | **71.3** | **70.0** | **77.4** | **76.5** | **68.0** |
| | red_cc | GPTAQ | **7.79** | 12.95 | **71.8** | **45.2** | 74.2 | 72.5 | 77.6 | 76.2 | 69.5 |
| | | GPTAQ+Ours | 7.85 | **12.71** | 69.6 | 44.4 | **74.5** | **73.0** | **79.6** | **77.2** | **69.7** |
| | c4 | GPTAQ | 8.39 | 12.96 | 72.1 | 45.0 | 70.1 | 71.0 | 77.9 | 76.9 | 68.8 |
| | | GPTAQ+Ours | **7.77** | **12.25** | **73.8** | **45.7** | **74.6** | **72.3** | **79.1** | **77.7** | **70.5** |
| | wiki2 | GPTAQ | 7.75 | 13.42 | 71.5 | 46.8 | **72.6** | 69.7 | 72.8 | 77.2 | 68.4 |
| | | GPTAQ+Ours | **7.43** | **12.80** | **73.2** | **47.4** | 68.7 | **73.0** | **77.0** | **78.0** | **69.6** |
| L3.1-8B -Instruct | - | FP16 | 7.21 | 11.39 | 79.6 | 54.8 | 79.1 | 74.1 | 83.9 | 80.9 | 75.4 |
| | red_stack | GPTAQ | **9.87** | 15.67 | 72.3 | 46.2 | **73.7** | **71.2** | 80.3 | 76.4 | 70.0 |
| | | GPTAQ+Ours | 9.89 | **15.55** | **75.7** | **49.5** | 72.8 | 68.3 | 80.3 | **77.8** | **70.7** |
| | red_cc | GPTAQ | 9.07 | 14.11 | 66.6 | 44.5 | 74.1 | **72.0** | **83.0** | 75.6 | 69.3 |
| | | GPTAQ+Ours | **8.98** | 14.11 | **76.2** | **48.4** | **74.3** | 71.5 | 82.8 | **77.1** | **71.7** |
| | c4 | GPTAQ | 8.90 | 13.85 | 70.5 | 46.0 | 74.5 | 71.7 | **81.5** | 76.9 | 70.3 |
| | | GPTAQ+Ours | **8.67** | **13.79** | **76.2** | **50.0** | **75.1** | **72.7** | 81.3 | **78.6** | **72.3** |
| | wiki2 | GPTAQ | 8.56 | **14.47** | 70.3 | 45.5 | 74.8 | 69.9 | 83.1 | 76.1 | 69.9 |
| | | GPTAQ+Ours | **8.18** | 14.56 | **73.6** | **48.0** | **74.9** | 71.5 | **83.6** | **77.2** | **71.4** |

selection. We validated our approach on three models using four distinct datasets: C4, WikiText-2, RedPajama (CommonCrawl subset), and RedPajama (StackExchange subset), while maintaining the 3-bit per-group weight-only quantization setting. The results are summarized in Table. 12. In most cases, our method demonstrates an improvement over GPTAQ, verifying its robustness to different calibration datasets.

## A.7 THE USE OF LLMS

In this paper, Large Language Models (LLMs) were used to assist with polishing the text and formatting the tables.

