# OpenReview forum: "Rethinking Residual Errors in Compensation-based LLM Quantization"
_ICLR.cc/2026/Conference — ICLR 2026 Poster_

### Official Review · Reviewer_mxfo · 2025-10-30

**Soundness:** 4
**Presentation:** 3
**Contribution:** 3
**Rating:** 6
**Confidence:** 3

**Summary:**

This paper proposes an improved residual error formulation for post-training quantization (PTQ) of large language models, building on frameworks like GPTQ and GPTAQ.

The key insight is that prior methods inaccurately model the calibration objective during the layer-wise iterative process. The authors identify that the residual error should not only account for the inter-layer propagated output error but also an intra-layer compensation-aware error arising from the weight update process itself.

**Strengths:**

This paper demonstrates significant strengths across key dimensions. Its originality lies in a nuanced reformulation of the residual error objective within established PTQ frameworks, identifying and incorporating the previously overlooked compensation-aware error.

The quality is high, evidenced by rigorous and extensive experiments across multiple model families (Llama 2/3), scales (1B-70B), and quantization settings (weight-only, weight-activation).

The clarity is commendable; the paper logically builds from the identified limitation to the proposed solution, with clear mathematical derivations and an efficiently described algorithm.

**Weaknesses:**

In significance, while improvements are consistent, they are often marginal (e.g., <1% accuracy gains in Table 1), raising questions about practical impact. The added calibration memory overhead (Table 5) could be prohibitive for edge devices, yet this trade-off is underexplored. Addressing these points would strengthen the work's relevance and applicability.

**Questions:**

Question 1 : Sensitivity to Calibration Data: Your experiments use a fixed 128 samples. How sensitive are the performance gains to the number and nature of the calibration samples? If the gains diminish significantly with fewer samples or change drastically with a different calibration dataset, it would impact the practical robustness of the method.

Question2 : Generalizability Beyond Llama Architectures: The empirical validation is comprehensive but exclusively on the Llama family. Have you observed similar improvements on other prominent architectures, such as GPT-style models (e.g., Qwen, Gemma) or encoder-only models? A result on at least one non-Llama model would greatly bolster the claim of general applicability.

---

> ### Author Response · Authors · 2025-11-21
> **Response to Reviewer mxfo**
>
> We sincerely thank the reviewer for the constructive comments, and we are encouraged by the appreciation of our work.
>
> ---
>
> **W1. The improvments are often marginal, raising questions about practical impact.**
>
> **WA1:** We wish to clarify that the seemingly 'modest' absolute gains in 3-bit settings are primarily due to performance saturation, as baselines are already approaching the floating-point (FP) upper bound. However, the value of our method becomes evident when considering the **relative reduction of the quantization error**. By effectively narrowing the performance gap between the quantized model and the FP baseline, our method recovers a significant portion of the remaining accuracy lost by previous methods.
> Furthermore, our method serves as a practical 'free lunch' for the quantization. It is **easy to deploy**, seamlessly integrating into existing frameworks like GPTQ and GPTAQ with negligible overhead (**~5% quantization time**) and **zero inference cost**. This makes it a robust, plug-and-play enhancement that offers practical impact.
>
> ---
>
> **W2. The added calibration memory overhead could be prohitive for edge devices**
>
> **WA2:** We sincerely apologize for your confusion. To the best of our knowledge, there are currently no edge devices capable of performing the GPTQ calibration process online. Calibration is typically conducted on a GPU to obtain the integer weights and scales, which are then downloaded to the edge device for inference. The additional calibration memory introduced by our method is strictly confined to the quantization and calibration phase on the GPU and does not involve adding any extra quantization parameters. When running on edge devices, our approach maintains the same integer weight format as GPTQ and GPTAQ, ensuring **no additional storage overhead or inference latency on edge devices**.
>
> ---
>
> **Q1. Senstivity to calibration data**
>
> **QA1:** Thank you for your valuable comment. First, we would like to clarify that the choice of 128 samples is the standard experimental configuration adopted by almost all compensation-based LLM quantization works, and was not an arbitrary selection made by us. To validate the robustness of our method, we further conduct experiments on
> (1) Dataset selection (3-bit per-group; C4 ppl/Avg Acc ↓/↑)
> |Model|Method|redpajama(stackexchange)|redpajama(commoncrawl)|c4|wiki2|
> |-----|------|---------|------|--|-----|
> |L2-7B|GPTAQ|8.75/65.6|8.56/66.5|8.38/66.6|9.22/66.0|
> ||GPTAQ+Ours|**8.72/66.8**|**8.33/66.6**|**8.19/67.6**|**8.56/67.0**|
> |L3-8B|GPTAQ|14.56/67.0|12.95/69.5|12.96/68.8|13.42/68.4|
> ||GPTAQ+Ours|**14.22/68.0**|**12.71/69.7**|**12.25/70.5**|**12.80/69.6**|
> |L3.1-8B-Ins|GPTAQ|15.67/70.0|14.11/69.3|13.85/70.3|14.47/69.9|
> ||GPTAQ+Ours|**15.55/70.7**|**14.05/71.7**|**13.79/72.3**|14.56/**71.4**|
>
> and (2) Number of samples (3-bit per-group; C4 ppl/Avg Acc ↓/↑)
> |Model|Method|64|96|128|192|
> |-----|------|--|--|---|---|
> |L2-7B|GPTAQ|8.44/66.5|8.42/66.4|8.40/66.3|8.38/66.6|
> ||GPTAQ+Ours|**8.31/66.6**|**8.23/66.9**|**8.19/66.6**|**8.19/67.6**|
> |L3.2-1B-Ins|GPTAQ|28.4/55.2|27.4/55.3|27.4/55.3|27.1/54.0|
> ||GPTAQ+Ours|**27.7/56.1**|**26.8/55.8**|**26.9/56.1**|**26.9/55.8**|
> |L3.1-8B|GPTAQ|12.59/71.4|12.82/70.7|12.26/71.0|12.32/71.5|
> ||GPTAQ+Ours|**12.37/71.7**|**12.21/71.4**|**12.08/71.6**|**12.06/71.8**|
> |L3-8B|GPTAQ|12.83/69.5|12.70/70.7|12.96/68.8|12.36/69.9|
> ||GPTAQ+Ours|**12.62/70.0**|**12.24/71.2**|**12.25/70.5**|**12.09/71.9**|
>
> Our proposed method achieves stable improvements, which demonstrates its robustness to the calibration set. The complete experimental results are also presented in Section A.6 of the revised PDF.
>
> ---
>
> **Q2: Generalizability beyond llama architectures**
>
> **QA2:** Thank you for the suggestion. Demonstrating generality is indeed important. We further conduct experiments on Qwen3 series as you suggestred, as demonstrated in the following table. We have also added these results in Sec.A.5 in the revised pdf.
> (2-bit per-group; C4 ppl/Avg Acc ↓/↑)
> |Method|Qwen3-4B|Qwen3-8B|Qwen3-14B|
> |------|-------|-------|--------|
> |GPTAQ (C4/ACC)|57.1/45.6|32.9/52.9|21.4/64.9|
> |GPTAQ+Ours (C4/ACC)|**39.9/52.2**|**29.9/54.8**|**20.9/65.1**|

---

### Official Review · Reviewer_CWPT · 2025-10-31

**Soundness:** 2
**Presentation:** 2
**Contribution:** 2
**Rating:** 4
**Confidence:** 4

**Summary:**

The paper shows that compensation-based PTQ methods for LLMs, such as GPTQ and especially GPTAQ, optimize against already compensated activations rather than the original full-precision (FP) layer output, which makes the target drift as we modify the weights inside the same layer. To fix this, the authors re-derive the per-layer objective so that every quantization and compensation step is directly aligned to the FP output, which naturally reveals a missing "compensation-aware" residual term capturing the error from earlier intra-layer updates, and they show that this term can be folded in using the same neuron-wise decomposition and Cholesky tools GPTAQ already uses, adding only modest memory/time overhead and delivering consistent quantization improvements for LLaMA 2/3 across especially in low-bit (3-2 bit) settings where quantization errors accumulate the most.

**Strengths:**

- The core insight is simple and just proposes that we should match to the original full-precision output instead of a compensated one.
- The derivation is intuitive because focusing on full-precision alignment, the missing compensation-aware term appears by default.
- The method easily fits within GPTAQ style pipelines since it also uses the neuron-wise and Cholesky.
- There are improvements in difficult low-bit and weight-plus-activation scenarios, which is exactly where we care.
- The experiments cover several LLaMA variants and quantization settings.

**Weaknesses:**

- Reported improvements are generally modest and in some cases the technique even leads to decreases in performance.
- The method introduces additional memory and runtime overhead, which is not always justified by the size of the gains.
- The writing of the paper has inconsistencies, e.g. Table 2 contains a bolding error on L3.1-8B-Inst for C4, where the proposed method is highlighted despite being worse than GPTAQ, which undermines the empirical presentation.
- Table 1 applies bolding only to the authors’ method even when other techniques perform better (e.g. L3-8B), creating presentation bias in favor of the proposed approach.
- The approach does not consistently dominate GPTAQ across models, on some models (again L3-8B) average accuracy even decreases in Table 1.

**Questions:**

- The evaluation is limited to LLaMA/LLaMA-3 variants. Evidence of the technique across other families (e.g. Qwen, Phi, Gemma), especially those with different activation/normalization patterns?
- It would be useful to extend Tables 5 and 6 to larger model to better understand the impact of the technique.
- How sensitive is the method to calibration set size and distribution?
- Are there quantization/deployment scenarios where aligning strictly to FP outputs is not the right target (e.g. fully quantized activation pipelines), and how would the method adapt there?
- Can the authors correct the bolding inconsistencies and re-run significance checks to ensure that the reported improvements are not due to formatting or selection bias?

I'd be happy to increase my score if the concerns are resolved!

---

> ### Author Response · Authors · 2025-11-21
> **Response to Reviewer CWPT**
>
> We sincerely thank the reviewer for the constructive comments, and we are encouraged by the appreciation of our work.
>
> ---
>
> **W1. reported improvements are generally modest and sometimes fail**
>
> **WA1:**   While baselines approach the floating-point (FP) upper bound at 3-bit settings, our method effectively narrows the remaining performance gap, significantly reducing the relative quantization error. Regarding accuracy fluctuations: Zero-shot Accuracy is a discrete metric susceptible to noise, whereas Perplexity (PPL) is a continuous, a more direct measure of quantization error. In the few cases where average accuracy drops, PPL consistently maintains a stable reduction, confirming the theoretical validity of our error correction.
>
> ---
>
> **W2. Additional memory and runtime overhead**
>
> **WA2:** The additional memory usage is strictly confined to the **calibration phase** and does not introduce extra quantization parameters. For inference, our approach maintains the identical integer weight format as GPTQ and GPTAQ, ensuring **no additional storage overhead or inference latency**.
>
> ---
>
> **W3&W4. Writing inconsistencies and bolding errors**
>
> **WA3&4:** We sincerely apologize for our oversight and thank you for pointing out the error. We have corrected the bolding mistake in the updated PDF, properly highlighting the methods with better metrics.
>
> ---
>
> **W5. Accuracy degradation on L3-8B**
>
> **WA5:** In Table 1, GPTAQ + Ours actually achieves a 1.7% improvement accuracy improvement over GPTAQ. I believe you may be referring to the accuracy decrease observed with **GPTQ + Ours**? Under the GPTQ + Ours setting on L3-8B, although accuracy slightly decreases, PPL (Perplexity) significantly improves, dropping from 8.53/13.28 to 8.0/12.53, significantly narrowing the gap between FP models. As we mentioned in **WA1** , PPL is a more direct indicator of quantization error compared to downstream task metrics, which validates that our calibration objective is more effective at reducing the quantization error relative to the floating-point model.
>
> ---
>
> **Q1. Generalizability beyond Llama**
>
>  **QA1:** Thank you for your valuable comment. We further conduct experiments on Qwen3 series, as demonstrated in the following table. We have also added these results in Sec.A.5 in the revised pdf.
> (2-bit per-group; C4 ppl/Avg Acc ↓/↑)
> |Method|Qwen3-4B|Qwen3-8B|Qwen3-14B|
> |------|-------|-------|--------|
> |GPTAQ (C4/ACC)|57.1/45.6|32.9/52.9|21.4/64.9|
> |GPTAQ+Ours (C4/ACC)|**39.9/52.2**|**29.9/54.8**|**20.9/65.1**|
>
> ---
>
> **Q2. Extend Tab.5 and Tab.6 to larger models**
>
> **QA2:** Thank you for your valuable comment. We have added **Llama3-70B** results to Tab.5 and Tab.6, and summarized the matrix need to perform calibation in Tab.10 in the revised PDF.
>
>  ---
>
>  **Q3. Calibration set ensitivity**
>  **QA3:** Thank you for your valuable comment. To validate the robustness of our method, we further conduct experiments on
>  (1) Dataset selection (3-bit per-group; C4 ppl/Avg Acc ↓/↑)
> |Model|Method|redpajama(stackexchange)|redpajama(commoncrawl)|c4|wiki2|
> |-----|-----|-----|-----|-----|-----|
> |L2-7B|GPTAQ|8.75/65.6|8.56/66.5|8.38/66.6|9.22/66.0|
> ||GPTAQ+Ours|**8.72/66.8**|**8.33/66.6**|**8.19/67.6**|**8.56/67.0**|
> |L3-8B|GPTAQ|14.56/67.0|12.95/69.5|12.96/68.8|13.42/68.4|
> ||GPTAQ+Ours|**14.22/68.0**|**12.71/69.7**|**12.25/70.5**|**12.80/69.6**|
> |L3.1-8B-Ins|GPTAQ|15.67/70.0|14.11/69.3|13.85/70.3|14.47/69.9|
> ||GPTAQ+Ours|**15.55/70.7**|**14.05/71.7**|**13.79/72.3**|14.56/**71.4**|
>
> and (2) Number of samples (3-bit per-group; C4 ppl/Avg Acc ↓/↑)
> |Model|Method|64|96|128|192|
> |-----|------|--|--|---|---|
> |L2-7B|GPTAQ|8.44/66.5|8.42/66.4|8.40/66.3|8.38/66.6|
> ||GPTAQ+Ours|**8.31/66.6**|**8.23/66.9**|**8.19/66.6**|**8.19/67.6**|
> |L3.2-1B-Ins|GPTAQ|28.4/55.2|27.4/55.3|27.4/55.3|27.1/54.0|
> ||GPTAQ+Ours|**27.7/56.1**|**26.8/55.8**|**26.9/56.1**|**26.9/55.8**|
> |L3.1-8B|GPTAQ|12.59/71.4|12.82/70.7|12.26/71.0|12.32/71.5|
> ||GPTAQ+Ours|**12.37/71.7**|**12.21/71.4**|**12.08/71.6**|**12.06/71.8**|
> |L3-8B|GPTAQ|12.83/69.5|12.70/70.7|12.96/68.8|12.36/69.9|
> ||GPTAQ+Ours|**12.62/70.0**|**12.24/71.2**|**12.25/70.5**|**12.09/71.9**|
>
> Our proposed method achieves stable improvements, which demonstrates its robustness to the calibration set. The complete experimental results are also presented in Section A.6 of the revised PDF.
>
>  ---
> **Q4. Still ailgn with FP output for fully activation quantized  pipelines?**
>
> **QA4:** Thank you for your valuable comment. Yes. GPTAQ adopts the order of **activation quantization first, then weight quantization**. This allows subsequent weight calibration to absorb errors introduced by activation quantization. Our method retains this same sequence, ensuring that error compensation remains effective and alignment with floating-point output is feasible even in fully quantized pipelines
>
>  ---
>
> **Q5. Correct bolding errors**
>
> **QA5:** Thank you for the helpful suggestion. We have correct those errors in the revised pdf.

---

> > ### Comment · Reviewer_CWPT · 2025-11-26
> > **Reviewer Response**
> >
> > Thank you for your answer, I've revised my score.

---

> > > ### Author Response · Authors · 2025-11-26
> > > **Thanks for your feedback**
> > >
> > > Dear Reviewer CWPT
> > >
> > > Thank your for your time and feedback! We sincerely appreciate your constructive reviews for improving our paper.
> > >
> > > Best regards,
> > >
> > > Authors of Paper #9296

---

### Official Review · Reviewer_xZwt · 2025-11-01

**Soundness:** 3
**Presentation:** 3
**Contribution:** 3
**Rating:** 2
**Confidence:** 5

**Summary:**

This paper presents a method that builds upon GPTQ and GPTAQ for efficient finetuning-free quantization. The main idea is to approximate the unquantized model’s activations using quantized weights during the quantization optimization process, aiming to reduce residual errors and improve quantization accuracy. The authors claim that this approach refines the calibration process and better aligns quantized activations with the full-precision model.

**Strengths:**

1. The topic of efficient and finetuning-free quantization for large transformer models is timely and relevant.

2. The authors provide a clear motivation to address accumulated quantization error across layers.

3. The structure of the paper and the technical presentation follow a standard quantization analysis format.

**Weaknesses:**

1. Sections 3.1 to 3.3 are almost entirely copied or rephrased from the GPTAQ paper. The derivations, notation, and even paragraph flow (e.g., the definitions of asymmetric calibration, residual error formulation, and inverse Hessian update) are reproduced with only superficial wording changes. This raises a serious concern of possible plagiarism.

2. The paper merely extends the asymmetric calibration idea already introduced in GPTAQ. The supposed “rethinking of residual error” is essentially a restatement of GPTAQ’s asymmetric calibration mechanism.

**Questions:**

1. Adding some figures might present the idea of this paper in a better way.
2. The content of this paper is not self-contained. Without reading the GPTAQ paper, it is hard to understand this paper. For example, it is hard to understand why in Eqn. 4, $\mathbf{\tilde{X}}$ is used as the target. The purpose of the background should completely explain the idea of the paper without copying content from previous papers. Concepts in previous papers should be explained in a concise way.

---

> ### Author Response · Authors · 2025-11-21
> **Response to Reviewer xZwt  (Part 1)**
>
> We sincerely thank the reviewer for the time and effort.
>
> ---
>
> **W1. background issues**
>
> **WA1:**
> We thank the reviewer for the detailed inspection of our background sections. We treat the concern regarding academic integrity with the utmost seriousness and appreciate the opportunity to clarify the intent and presentation of these sections.
>
> **(1) Clarification on Attribution (Not Plagiarism)** We respectfully clarify that Sections 3.1–3.3 were explicitly intended as a review of preliminaries to establish the problem context, not as our own contribution. As verified in our original submission:
>
> -   **Section 3.2** explicitly cites **OBQ [Frantar & Alistarh, 2022]** in the first sentence, and cites **GPTQ(Frantar et al., 2022)** before introducing its innovations.
>
> -   **Section 3.3** explicitly cites **GPTAQ [Li et al., 2025]** in the first sentence. The text clearly attributes the methods and formulas to these prior works.
>
> **(2) Reason for Similarity & Acknowledgement.** The textual similarity arose because we deliberately adhered to the **exact mathematical notation and derivation flow** of the baselines.
> - **Consistency:** Since our proposed method is a mathematical refinement of the error modeling in these specific works, we prioritized notation consistency to help readers directly compare our contribution (Section 4) against the standard baseline.
> - **Acknowledgement:** However, we acknowledge that the description in the submitted version tracked the original text too closely. We agree with the reviewer that this similarity—even with citations—was not ideal for presentation. Furthermore, we realize that this similarity may have inadvertently obscured the distinct mathematical refinements we introduce in Section 4, blurring the line between the standard background and our novel contributions.
>
> **(3) Revisions in the Updated PDF.** To address this concern comprehensively, we have **significantly revised** the background sections in the uploaded PDF:
>
> -   **Rewritten Background:** We have completely rewritten Sections 3.1–3.3. The descriptions are now condensed and synthesized. We focus on articulating the process by which GPTQ and GPTAQ translate the layer-level objective into the column-wise objective function and the compensation term derivation.
>
> - **Explicit Clarification:** We added an explicit statement: _"We adopt the problem formulation and notation from [Li et al., 2025] solely to facilitate direct comparison."_ reinforcing the citations already present.
>
> This revision makes the background concise and clearly separates prior arts from our novel contributions.

---

> ### Author Response · Authors · 2025-11-21
> **Response to Reviewer xZwt (Part2)**
>
> **W2. Novelty compared to GPTAQ**
>
> **WA2:**  We wish to clarify the fundamental difference and novelty compared to GPTAQ
> 1. Conceptual Difference: Inter-layer vs. Intra-layer Error
> We would like to clarify that GPTAQ and our method address two fundamentally different sources of error:
>
> -   **GPTAQ addresses the Input Error (Inter-layer):** GPTAQ observes that the input to the current layer changes from $\tilde{X}$ (FP flow) to $X$ (Quant flow) due to quantization in previous layers. It introduces a residual term to align the input flow.
>
> -   **Our Method addresses the Compensation Error (Intra-layer):** We identify a new error source _within_ the layer’s iterative quantization process. Even if the input were perfect, standard block-wise quantization (like GPTQ/GPTAQ) updates the remaining FP weights ($W^{(q)}$) to compensate for quantization errors. We prove that previous methods incorrectly use the output of these _updated_ weights ($W^{(q)}\tilde{X}$) as the calibration target. Our method corrects this by strictly aligning with the _original_ FP weights ($W^{(0)}\tilde{X}$), introducing the "Compensation-Aware Error" term.
>
>
> 2. Mathematical Distinctness
> The distinction is mathematically rigorous. The residual error in our paper is reformulated as:
>
> $$
> r' = \underbrace{w^{(q)}(\tilde{X} - X)} _{ r1: \text{GPTAQ term} } + \underbrace{(w^{(0)} - w^{(q)})\tilde{X}} _{ r2: \text{Ours (New term)} }
> $$
>
> -   The first term ($r1$) accounts for the input discrepancy, which is indeed the contribution of GPTAQ.
>
> -   The second term ($r2$), which we derive, accounts for the discrepancy between the original weights and the intermediate compensated weights. This term is absent in GPTAQ. Therefore, our formulation is not a restatement but a **mathematically strictly more accurate generalization** that encompasses GPTAQ as a special case (where $r2$ is ignored).
>
> 3. Empirical Proof of Independence
> our **Ablation Study (Table 4)**  explicitly contradicts this. Even if we sololy uses the $r2$ term, GPTQ + Ours outperforms standard GPTQ.
>
> In summary, while we build upon the efficient decomposition technique used in GPTAQ, our contribution is the identification and correction of a distinct, previously overlooked "Compensation-Aware Error." This leads to a more precise calibration objective and significant performance gains.
>
> ---
>
> **Q1. Add figures for better presentation**
>
> **QA1:**
> Thank you for your valuable comment. We have added **Figure 1** in the revised PDF to illustrate the fundamental quantization pipeline of **GPTQ, GPTAQ, and Ours**, and to clearly demonstrate the difference between our method and these baselines.
>
> ---
>
> **Q2. Better explain preivious works and core idea**
>
> **QA2:** We sincerely apologize for the difficulty you encountered in reading our manuscript. In the updated PDF, we have **restructured the background and parts of the methodology section** to enhance clarity. Our revised structure now proceeds as follows:
>
> 1.  We first explain the **differences in input sources** utilized by GPTQ and GPTAQ.
>
> 2.  We then articulate how both GPTQ and GPTAQ transition from the **high-level (layer-level) optimization objective** to formulating the objective function for **quantizing each column**, which ultimately leads to the derivation of the **compensation term** calculation.
>
> 3.  Building upon this foundation, we analyze why the column-level optimization objective in GPTAQ is **inaccurate**, which **logically leads to the introduction of our proposed method.**

---

### Official Review · Reviewer_shR7 · 2025-11-01

**Soundness:** 3
**Presentation:** 3
**Contribution:** 4
**Rating:** 8
**Confidence:** 5

**Summary:**

This paper points out a defective implementation in the previously published GPTAQ method, proposes a corrected algorithm, and demonstrated the effectiveness of the correction.

**Strengths:**

+ Clear presentation.
+ Sound analysis.
+ Compelling empirical results.
+ Practical significance.

**Weaknesses:**

- I have but one issue with the wording when compared against GPTAQ.  If I understand correctly, the formulation of GPTAQ optimization in matrix form is exactly correct and not challenged by this paper; rather, the row-wise iterative algorithm implementation has been defective and is now corrected to truly align with the optimization problem.  So instead of leaving the reader the impression of this being yet another method, it should be clearly shown as a correction to a previously wrongly implemented existing method, which is no less significant.

**Questions:**

* Minor question on data-efficiency: to achieve the same compensation outcome, does the corrected algorithm require more data than the original GPTQ/GPTAQ?

---

> ### Author Response · Authors · 2025-11-21
> **Response to Reviewer shR7**
>
> We sincerely thank the reviewer for the constructive comments, and we are encouraged by the appreciation of our work.
>
> ---
>
> **W1. Clarify the comparison with GPTAQ**
>
> **WA1:** Thank you for your valuable comment. You are correct that the **high-level (layer-level) optimization objective** of GPTAQ is sound; it attempts to align the output of the quantized model with that of the original floating-point model at every layer, by using the actual floating-point layer output $\mathbf{w} \widetilde{\mathbf{X}}$ as the calibration reference.
>
> $$\min_{\widehat{\mathbf{w}}} \left\| \widehat{\mathbf{w}} \mathbf{X} - \mathbf{w} \widetilde{\mathbf{X}} \right\|_F^2$$
>
> However, at the **column level**, GPTAQ incorrectly uses the output of these _updated_ weights ($W^{(q)}\tilde{X}$) as the calibration target. Our method corrects this by strictly aligning with the _original_ FP (floating-point) weights ($W^{(0)}\tilde{X}$), introducing the “**Compensation-Aware Error**” term.
>
> In the updated PDF, we have restructured the background and parts of the methodology to better explain that **while the high-level objective proposed by GPTAQ is correct, it is distorted at the column level**, and our contribution is precisely to enforce alignment with the floating-point output even at the column level.
>
> ---
>
> **Q1. data-efficiency issues**
>
> **QA1:** Thank you for your valuable comment. Across all experiments in the main text, we utilized **128 samples** for calibration, which is **entirely consistent with the settings used in GPTQ and GPTAQ**. Furthermore, we have included supplementary experiments comparing performance under a varying number of samples, as shown in the table below. Our method demonstrates considerable stability even with different numbers of calibration samples. These supplementary results have also been added to Section A.6 of the updated PDF.
>
> |Model|Method|64|96|128|192|
> |-----|------|--|--|---|---|
> |L2-7B|GPTAQ|8.44/66.5|8.42/66.4|8.40/66.3|8.38/66.6|
> ||GPTAQ+Ours|**8.31/66.6**|**8.23/66.9**|**8.19/66.6**|**8.19/67.6**|
> |L3.2-1B-Ins|GPTAQ|28.4/55.2|27.4/55.3|27.4/55.3|27.1/54.0|
> ||GPTAQ+Ours|**27.7/56.1**|**26.8/55.8**|**26.9/56.1**|**26.9/55.8**|
> |L3.1-8B|GPTAQ|12.59/71.4|12.82/70.7|12.26/71.0|12.32/71.5|
> ||GPTAQ+Ours|**12.37/71.7**|**12.21/71.4**|**12.08/71.6**|**12.06/71.8**|
> |L3-8B|GPTAQ|12.83/69.5|12.70/70.7|12.96/68.8|12.36/69.9|
> ||GPTAQ+Ours|**12.62/70.0**|**12.24/71.2**|**12.25/70.5**|**12.09/71.9**|

---

### Author Response · Authors · 2025-12-02
**Rebuttal Summary of Submission #9296**

Dear Area Chair,

We would like to express our sincere gratitude for you and reviewers for the constructive feedback and the time dedicated to evaluating our work. We are encouraged that all reviewers consistently recognized the **clear motivation** (shR7, xZwt, CWPT, mxfo) of our work in identifying the overlooked compensation-aware error. They also praised the **clear mathematical derivations** (shR7, CWPT, mxfo) and the **extensive experiments** (shR7, CWPT, mxfo). Notably, **Reviewer CWPT raised their score from 4 to 6** following the additional validation provided during the rebuttal.

Although we regret that other reviewers did not engage in further discussion during rebuttal, we are confident that we have resolved their primary concerns regarding background presentation, generalizability & robustness, and calibration overhead with extensive additional experiments and clarifications. Below is the summary.

### 1.  Addressing the Presentation Concern (Response to Reviewer xZwt)

-   **Firm Clarification on Integrity:** We treat the concern regarding academic integrity with the utmost seriousness. We respectfully but firmly clarify that this was **an issue of presentation, not plagiarism or academic misconduct.** In our original submission, these sections **explicitly cited the baseline papers (OBQ/GPTQ/GPTAQ)**. We adopted the exact notation and derivation flow strictly only to facilitate a direct mathematical comparison and to pinpoint where our correction diverges from their formulation.

-   **Action Taken:** However, we acknowledge that the description tracking too closely—even with citations—was not appropriate. **In the revised PDF, we have fully rewritten Sections 3.1–3.3.** The new text is condensed, synthesized, and clearly distinguishes prior arts from our novel contributions. This complete rewriting eliminates any ambiguity and resolves the presentation concern. We also added Figure 1 to visually illustrate the structural difference between our pipeline and GPTQ/GPTAQ.

### 2. Clarifying Novelty & Contribution (Response to Reviewer xZwt)
-   **Mathematical Distinction:** We clarified that we identify and fix the **Intra-layer** compensation error (unlike GPTAQ's Inter-layer focus). We provided a rigorous **mathematical proof** showing our formulation is a strictly more accurate generalization.

-   **Consensus:** Reviewer shR7 praised this as "Sound analysis," Reviewer CWPT praised our "derivation is intuitive", and Reviewer mxfo acknowledged the "Originality lies in a nuanced reformulation."

### 3. Enhanced Generalizability & Robustness (Response to Reviewer CWPT and mxfo)

Both Reviewers requested validation beyond the Llama family and sensitivity analysis. We have added extensive experiments:

-   **New Models:** We extended evaluation to **Qwen3 (4B, 8B, 14B)**. Results (added to Appendix A.5) show consistent improvements across different architectures.

-   **Robustness:** We conducted sensitivity analyses on different **calibration datasets** (RedPajama, Wiki2, C4) and **sample sizes** (64-192). Results (added to Appendix A.6) confirm our method is robust to data variations.

### 4. Clarification on Overhead (Response to Reviewer CWPT and mxfo)

We clarified that the memory overhead is strictly confined to the **offline calibration phase (on GPU)**. The inference process remains identical to standard integer-only inference, incurring **zero overhead** on edge devices.

### **Conclusion**


Our paper provides a mathematically sound correction to the calibration objective of LLM quantization. With the background section completely rewritten and the generalizability & robustness validation strengthened, we believe the reviewers' concerns can be fully addressed. We again thank your time and effort.

Best regards,

Authors of Paper #9296

---

> ### Comment · Area_Chair_Jqy1 · 2025-12-03
>
> Thank you for summarizing the review comments and the rebuttal. I will review everything carefully.
> Your new AC

---

### Meta-Review · Area_Chair_Jqy1 · 2026-01-04

**Summary:**

Reviewer shR7’s main concern was not about technical correctness but about framing and positioning. While acknowledging the sound analysis and strong empirical results, this reviewer felt that the paper could be misinterpreted as proposing an entirely new method, whereas it is more accurately a correction of a defective column-wise implementation in GPTAQ. The reviewer emphasized that this distinction should be made explicit to avoid overstating novelty. In addition, the reviewer asked whether the corrected formulation required more calibration data than GPTQ or GPTAQ, raising a question about data efficiency.

Reviewer xZwt focused primarily on presentation and originality. This reviewer expressed serious concern that Sections 3.1–3.3 were too close to the GPTAQ paper in structure, notation, and derivation flow, to the point of raising a potential plagiarism concern. They also questioned whether the proposed “rethinking of residual error” was substantively different from GPTAQ’s asymmetric calibration or largely a restatement. Beyond originality, this reviewer found the paper insufficiently self-contained, noting that key equations and design choices were difficult to understand without reading GPTAQ, and suggested adding figures and clearer background explanations.

Reviewer CWPT raised concerns about the practical impact and consistency of the results. Although they found the core insight intuitive and the derivation reasonable, they noted that the reported improvements were generally modest and sometimes inconsistent, with occasional performance regressions. They also questioned whether the additional calibration memory and runtime overhead were justified by the gains, and highlighted presentation issues such as incorrect or biased bolding in tables that undermined confidence in the empirical claims. This reviewer further asked for broader validation beyond LLaMA models, sensitivity analyses with respect to calibration data size and distribution, and clarification on whether strict alignment to full-precision outputs remains appropriate in fully quantized activation pipelines.

Reviewer mxfo was generally positive about the technical contribution but raised concerns similar in spirit to CWPT regarding practical significance. While acknowledging the originality of the compensation-aware residual formulation and the rigor of the experiments, they noted that the absolute gains were often small and questioned the real-world impact. They also pointed out that the added calibration memory overhead could be problematic for edge deployment scenarios if not carefully contextualized. Finally, they requested stronger evidence of robustness to calibration data choices and generalizability beyond the LLaMA family, suggesting that results on at least one non-LLaMA architecture would strengthen the paper.

Together, these reviewer-specific concerns informed the overall assessment by highlighting issues of framing and originality, clarity and self-containment, empirical consistency and significance, deployment overhead, and generalizability, even though reviewers broadly agreed on the relevance of the problem and the soundness of the core technical idea.

**Reviewer Concerns:**

This meta-reviewer believes that the rebuttal addressed the majority of the substantive concerns raised during the review process, with only minor residual issues remaining.

Most importantly, the concern raised by Reviewer xZwt regarding the paper’s similarity to GPTAQ, including the risk of being perceived as a restatement or even potential plagiarism, was directly and seriously addressed. In the rebuttal, the authors clearly clarified that the overlapping sections were intended purely as background and preliminaries, explicitly attributed to prior work, and that the similarity stemmed from a deliberate choice to preserve notation and derivation flow for precise mathematical comparison. More critically, the authors went beyond a textual defense: they fully rewrote the background sections, explicitly separated prior work from their own contribution, added clarifying statements on adopted formulations, and introduced a new figure to visually distinguish their pipeline from GPTQ and GPTAQ. From this meta-reviewer’s perspective, these actions substantially mitigate the originality and presentation concerns and make the distinction between the proposed method and GPTAQ much clearer.

On the technical side, the rebuttal convincingly clarified novelty by articulating a clean conceptual and mathematical separation between GPTAQ’s inter-layer input error and the newly identified intra-layer compensation-aware error. The mathematical reformulation and ablation evidence presented in the rebuttal support the claim that the proposed method is not a restatement but a strictly more accurate generalization. This directly addresses Reviewer xZwt’s skepticism as well as related concerns from other reviewers.

Concerns about generalizability and robustness, raised by Reviewers CWPT and mxfo, were also largely resolved. The authors added new experiments on non-LLaMA models, conducted extensive sensitivity analyses across calibration datasets and sample sizes, and demonstrated stable improvements. These additions meaningfully strengthen confidence in the method’s applicability beyond the originally evaluated settings. Similarly, questions about memory and runtime overhead were clarified by explicitly distinguishing offline calibration costs from inference-time deployment, resolving confusion about practical deployment impact.

The remaining outstanding concerns are relatively minor and mostly relate to the modest magnitude of the reported gains and their variability across models. While the rebuttal provides a reasonable explanation—performance saturation near FP baselines and the use of perplexity as a more faithful error metric—some reviewers may still view the absolute improvements as limited. However, this meta-reviewer considers these to be judgment calls about impact rather than unresolved technical flaws.

Overall, this meta-reviewer finds that the rebuttal substantially strengthens the paper, resolves the most serious concerns—especially the similarity-to-GPTAQ issue raised by Reviewer xZwt—and clarifies both novelty and scope. In light of the clarified contribution, corrected presentation, and expanded experimental validation, the remaining issues do not outweigh the paper’s technical soundness and relevance, and the overall assessment leans toward acceptance.

**Reviewer Scores:**

This meta-reviewer anticipates the following score changes had each reviewer been able to participate fully in the post-rebuttal discussion.

For Reviewer CWPT, who initially assigned a score of 4, the reviewer explicitly stated during the discussion that they had raised their score after reviewing the rebuttal and the additional experimental evidence. This confirms that their primary concerns regarding empirical consistency, presentation issues, overhead clarification, and broader validation were satisfactorily addressed by the authors.

For Reviewer xZwt, who initially gave a score of 2, this meta-reviewer believes the score could reasonably have increased had the reviewer participated fully in the discussion. The reviewer’s main concern centered on the perceived excessive similarity to GPTAQ and doubts about originality. The rebuttal addressed this issue directly by clearly articulating the conceptual and mathematical distinction between GPTAQ’s inter-layer error and the newly identified intra-layer compensation-aware error, demonstrating that the proposed formulation is a strict generalization rather than a restatement, and fully rewriting the background sections to clearly separate prior work from novel contributions. From this meta-reviewer’s perspective, these responses substantially resolve the core GPTAQ-related concern that motivated the low initial score.

For Reviewer shR7, who originally assigned a score of 8, the assessment was already strongly positive, emphasizing sound analysis, clear presentation, and practical significance. The reviewer’s concerns were limited to framing and wording, particularly how the contribution should be positioned relative to GPTAQ. The rebuttal directly clarified this positioning and confirmed unchanged data efficiency, which is consistent with maintaining a high score.

For Reviewer mxfo, who gave a score of 6, the review was generally supportive, with remaining concerns focused on the modest magnitude of gains, calibration overhead, and generalizability. The rebuttal addressed these points through additional experiments, robustness analyses, and clearer discussion of deployment overhead. These responses further strengthen the paper but are unlikely to materially change an already moderately positive assessment.

Overall, this meta-reviewer believes that the rebuttal effectively resolved the main points of contention, led to a confirmed score increase from one reviewer, and substantially clarified the most critical remaining concern regarding similarity to GPTAQ, supporting a favorable overall evaluation of the paper.

---

### Decision · Program_Chairs · 2026-01-26

Accept (Poster)